# Hammerhead-type FXR agonists induce an enhancer RNA *Fincor* that ameliorates nonalcoholic steatohepatitis in mice

Jinjing Chen[1†], Ruoyu Wang[2†], Feng Xiong[2], Hao Sun[1], Byron Kemper[1], Wenbo Li[2*], Jongsook Kemper[1*]

[1]Department of Molecular and Integrative Physiology, University of Illinois at Urbana-Champaign, Urbana, United States; [2]Department of Biochemistry and Molecular Biology, McGovern Medical School, University of Texas Health Science Center, Houston, United States

*For correspondence:
wenbo.li@uth.tmc.edu (WL);
jongsook@illinois.edu (JK)

[†]These authors contributed equally to this work

**Competing interest:** The authors declare that no competing interests exist.

**Abstract** The nuclear receptor, farnesoid X receptor (FXR/NR1H4), is increasingly recognized as a promising drug target for metabolic diseases, including nonalcoholic steatohepatitis (NASH). Protein-coding genes regulated by FXR are well known, but whether FXR also acts through regulation of long non-coding RNAs (lncRNAs), which vastly outnumber protein-coding genes, remains unknown. Utilizing RNA-seq and global run-on sequencing (GRO-seq) analyses in mouse liver, we found that FXR activation affects the expression of many RNA transcripts from chromatin regions bearing enhancer features. Among these we discovered a previously unannotated liver-enriched enhancer-derived lncRNA (eRNA), termed FXR-induced non-coding RNA (*Fincor*). We show that *Fincor* is specifically induced by the hammerhead-type FXR agonists, including GW4064 and tropifexor. CRISPR/Cas9-mediated liver-specific knockdown of *Fincor* in dietary NASH mice reduced the beneficial effects of tropifexor, an FXR agonist currently in clinical trials for NASH and primary biliary cholangitis (PBC), indicating that amelioration of liver fibrosis and inflammation in NASH treatment by tropifexor is mediated in part by *Fincor*. Overall, our findings highlight that pharmacological activation of FXR by hammerhead-type agonists induces a novel eRNA, *Fincor*, contributing to the amelioration of NASH in mice. *Fincor* may represent a new drug target for addressing metabolic disorders, including NASH.

## eLife assessment

Using unbiased transcriptional profiling, the study reports a **fundamental** discovery of FincoR, a novel hepatic lncRNA generated from an enhancer element, which plays a role in FXR biology. The **convincing** findings have therapeutic implications in the treatment of MASH. The authors use state-of-the-art methodology and use unbiased transcriptomic profiling and epigenetic profiling, including validation in mouse models and human samples.

## Introduction

Nonalcoholic fatty liver disease (NAFLD) is the most common chronic liver disease and a leading cause of liver transplants and liver-related death (*Friedman et al., 2018*). NAFLD begins with simple steatosis but may further progress to a severe form, nonalcoholic steatohepatitis (NASH), and later, fatal cirrhosis and liver cancer (*Friedman et al., 2018*). Despite its striking global increase and clinical importance, there is no approved drug for NASH. The urgent need for development of therapeutic

**eLife digest** Non-alcoholic steatohepatitis, also known as NASH, is a severe condition whereby fat deposits around the liver lead to inflammation, swelling, scarring and lasting damage to the organ. Despite being one of the leading causes of liver-related deaths worldwide, the disease has no approved treatment.

A protein known as Farnesoid X receptor (or FXR) is increasingly being recognized as a promising drug target for non-alcoholic steatohepatitis. Once activated, FXR helps to regulate the activity of DNA regions which are coding for proteins important for liver health. However, less is known about how FXR may act on non-coding regions, the DNA sequences that do not generate proteins but can be transcribed into RNA molecules with important biological roles.

In response, Chen et al. investigated whether FXR activation of non-coding RNAs could be linked to the clinical benefits of hammerhead FXR agonists, a type of synthetic compounds that activates this receptor.

To do so, genetic analyses of mouse livers were performed to identify non-coding RNAs generated when FXR was activated by the agonist. These experiments revealed that agonist-activated FXR induced a range of non-coding RNAs transcribed from DNA sequences known as enhancers, which help to regulate gene expression. In particular, hammerhead FXR agonists led to the production of a liver-specific enhancer RNA called *Fincor*.

Additional experiments using tropifexor, a hammerhead FXR agonist currently into clinical trials, showed that this investigational new drug had reduced benefits in a mouse model of non-alcoholic steatohepatitis with low *Fincor* levels. This suggested that this enhancer RNA may play a key role in mediating the clinical benefits of hammerhead FXR agonists, encouraging further research into its role and therapeutic value.

agents for NASH has greatly increased research interest in the nuclear receptor, farnesoid X receptor (FXR, NR1H4) (*Evans and Mangelsdorf, 2014*).

FXR is activated by its physiological ligands, bile acids (BAs), and regulates expression of genes involved in BA, lipid, and glucose metabolism and hepatic autophagy, which maintain metabolite levels and metabolic homeostasis (*Calkin and Tontonoz, 2012*; *Kliewer and Mangelsdorf, 2015*; *Lee et al., 2006*; *Lee et al., 2014*; *Seok et al., 2014*). Ligand-activated FXR also protects against hepatic inflammation and liver injury (*Jung et al., 2020*; *Wang et al., 2008*). The action of FXR, similar to other nuclear receptors, is achieved primarily by its binding to chromatin loci to regulate the transcription of target genes (*Calkin and Tontonoz, 2012*; *Lee et al., 2006*; *Lee et al., 2012*; *Thomas et al., 2010*). Consistent with its crucial physiological functions, FXR is increasingly recognized as a promising drug target, particularly for liver diseases, such as NASH and primary biliary cholangitis (PBC) (*Abenavoli et al., 2018*; *Ali et al., 2015*; *Downes et al., 2003*; *Kremoser, 2021*). For example, semi-synthetic or non-steroidal synthetic agonists of FXR, including obeticholic acid (OCA) and hammerhead-type agonists, such as tropifexor and cilofexor, are currently in clinical trials for NASH and PBC patients (*Abenavoli et al., 2018*; *Kremoser, 2021*; *Sanyal et al., 2023*; *Tully et al., 2017*). However, how pharmacological activation of FXR mediates such beneficial therapeutic effects is poorly understood.

Non-protein-coding RNAs (ncRNAs) are one of the fascinating discoveries of modern biology (*Cech and Steitz, 2014*). While a significant portion of the genome was initially thought to be 'junk DNA', it has been established that many non-coding regions give rise to functional non-coding RNAs (ncRNAs) (*Cech and Steitz, 2014*). Of these ncRNAs, long non-coding RNAs (lncRNAs) are a group of transcripts longer than 200 nucleotides and play important roles in diverse biological processes (*Lam et al., 2014*; *Li et al., 2016*; *Mattick et al., 2023*; *Sartorelli and Lauberth, 2020*; *Statello et al., 2021*). A group of lncRNAs are produced from genomic regions bearing epigenetic features of enhancers (*Hon et al., 2017*; *Mattick et al., 2023*). This is consistent with the idea that many transcriptional enhancers actively transcribe ncRNAs that are referred to as enhancer RNAs (eRNAs, >85k in humans and >57k in mice) (*Hirabayashi et al., 2019*), some of which are functionally important for enhancer functions (*Lai and Shiekhattar, 2014*; *Li et al., 2016*; *Sartorelli and Lauberth, 2020*). A majority of eRNAs have not yet been annotated in current lncRNA databases, such as GENCODE. Exploring eRNA landscapes and functions in diverse biology and disease states will facilitate our

understanding of both lncRNAs and enhancers (*Li et al., 2016*; *Mattick et al., 2023*). Indeed, the landscape of eRNAs in mouse liver has been minimally explored (*Fang et al., 2014*), and has not been well studied in response to specific nuclear receptor activation, such as by FXR. Moreover, functional roles of eRNAs in vivo in intact organisms is understudied.

FXR was shown to regulate expression of small non-coding microRNA (miR) genes, e.g., *Mir34a and Mir802* (*Lee et al., 2010*; *Seok et al., 2021*), but it has not been reported whether FXR achieves its function through regulating lncRNAs, which far outnumber miRs. Because FXR directly regulates expression of its target genes (*Lee et al., 2012*; *Thomas et al., 2010*), we examined if FXR regulates eRNAs, and if such transcripts participate in its physiological or pharmacological functions. By utilizing RNA-seq and global run-on sequencing (GRO-seq) analyses of livers from mice treated with FXR ligands, we identified a set of FXR-regulated eRNAs. Among these, we focused on a highly induced and abundantly expressed eRNA that we referred to as FXR-induced non-coding RNA (*Fincor*) for functional studies. *Fincor* is highly enriched in mouse liver and is induced specifically by hammerhead-type FXR agonists, including GW4064 and tropifexor. In vivo studies utilizing CRISPR/Cas9-mediated liver-specific knockdown of *Fincor* in dietary NASH mice indicated that *Fincor* is critically involved in mediating the beneficial pharmacological effects of tropifexor in reducing liver fibrosis and inflammation.

## Results

### Activation of FXR by GW4064 induces a novel eRNA, *Fincor*, in mouse liver

To identify eRNAs potentially regulated by FXR, we first obtained a list of putative enhancers in mouse liver based on ENCODE H3K27ac ChIP-seq data (ENCFF001KMI, see Materials and methods). Then, we performed ribo-depleted total RNA-seq in the livers from mice treated with a specific FXR agonist, GW4064, to identify transcripts produced from these enhancer regions (see *Supplementary file 1a*). To avoid confounding issues of RNA signals from genes, we specifically focused on intergenic enhancer regions (±3 kb from H3K27ac peak center) that harbor discernible RNA-seq signals (RPKM >1).

This genomic analysis resulted in identification of 190 high-confidence eRNAs in mouse liver. Among these, 14 eRNAs were upregulated and 5 were downregulated by GW4064 treatment (FDR <0.05, $\log_2$FC >1, *Figure 1A*) (see *Supplementary file 1b*). FXR-regulated eRNAs were produced adjacent to many genes with important roles in liver metabolism and disease, e.g., *Hes1* (*Figure 1—figure supplement 1A and B*). One of the most robustly induced eRNAs was *Fincor*, an unannotated novel transcript located on chromosome 19 (*Figure 1A, B, and C*). Reverse transcription qPCR (RT-qPCR) confirmed that treatment with GW4064 substantially induced expression of *Fincor,* more than 10-fold in mouse liver, which is similar to induction of *Nr0b2*, a well-known FXR target gene (*Figure 1D*; *Claudel et al., 2005*; *Evans and Mangelsdorf, 2014*; *Goodwin et al., 2000*). Induction of *Fincor* by GW4064 was transient, peaked within 1 hr and then declined gradually, a pattern similar to that of *Nr0b2* (*Figure 1—figure supplement 1C*). Expression of genes adjacent to *Fincor*, including *Gcnt1, Rfk, Pcsk5, and Prune2,* did not change after acute 1 hr FXR activation as shown by RNA-seq (*Figure 1C*), and confirmed for *Gcnt1* by time course qPCR (*Figure 1—figure supplement 1C*).

We also found that the short-time GW4064 treatment resulted in 590 upregulated genes and 500 downregulated genes (FDR <0.05) (*Figure 1—figure supplement 2A*). Gene ontology (GO) enrichment analysis of these differentially expressed genes revealed their roles in the regulation of triglyceride, fatty acid, and cholesterol metabolism (*Figure 1—figure supplement 2B*), which is consistent with known roles of FXR in these physiological processes (*Claudel et al., 2005*).

### Ligand-activated FXR directly activates transcription of eRNAs, including *Fincor*

We sought to examine: (1) if these eRNAs were directly activated by FXR, and (2) if the activation takes place transcriptionally using *Fxr* liver-specific knockout (*Fxr-LKO*) mice that were treated with GW4064 to determine if FXR was required for induction of eRNAs by GW4064. *Fxr-LKO* mice were generated from *Fxr* floxed (*Fxr-Flox*) mice (*Figure 2—figure supplement 1A*) and transcription of eRNAs was detected by GRO-seq, a widely used method to detect nascent RNA transcription, including eRNAs (*Lam et al., 2014*; *Li et al., 2016*; *Sartorelli and Lauberth, 2020*). GRO-seq showed that FXR-induced eRNAs were activated transcriptionally by GW4064 in *Fxr-flox* mice, but such induction was abolished

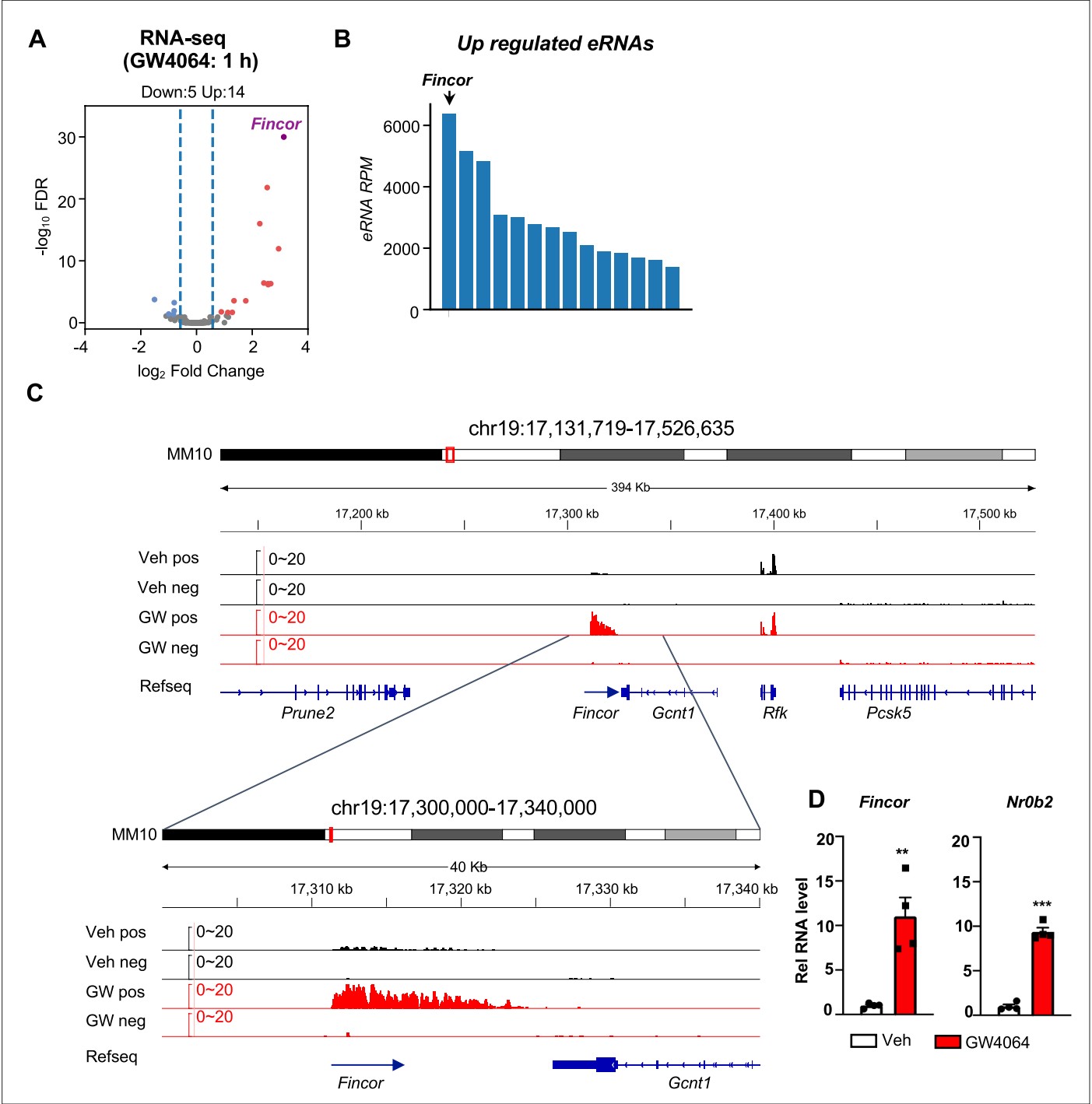

**Figure 1.** Activation of farnesoid X receptor (FXR) by GW4064 induces FXR-induced non-coding RNA (*Fincor*), a novel enhancer RNA (eRNA), in mouse liver. (**A**) Volcano plot from RNA-seq showing significantly induced eRNAs (*Fincor* is highlighted) in the livers of C57BL/6 male mice treated with GW4064 (i.p. injection, 30 mg/kg, 1 hr) or vehicle. The x axis denotes $\log_2$ fold change (GW4064/Veh) of eRNAs and the y axis denotes -$\log_{10}$ FDR of eRNAs. (**B**) A bar plot showing the reads per million (RPMs) of upregulated eRNAs by GW4064 treatment. (**C**) IGV genome browser track showing RNA-seq signals from vehicle or GW4064-treated samples around the *Fincor* locus and its neighboring regions. A zoom-in view of *Fincor* is shown below. Veh, vehicle; GW, GW4064; pos, positive strand; neg, negative strand; Refseq, reference sequence. (**D**) Reverse transcription qPCR (RT-qPCR) data showing GW4064 induction of *Fincor* eRNA and *Nr0b2* mRNA in the liver (n=4/group). Data are presented as mean ± SEM. Statistical significance was determined by the two-way ANOVA Sidak's multiple comparisons test with **p<0.01 and ***p<0.001.

The online version of this article includes the following figure supplement(s) for figure 1:

**Figure supplement 1.** Examples of FXR-regulated eRNAs and time course expression of Fincor after FXR activation.

**Figure supplement 2.** RNA-seq analysis of the differentially expressed genes in the GW4064-treated mouse liver.

in *Fxr-LKO* mice (*Figure 2A*). In particular, *Fincor* is robustly induced in the GRO-seq analysis and its induction is dependent on hepatic FXR (*Figure 2B*). RT-qPCR confirmed the FXR-dependent expression of *Fincor*, similar to that of *Nr0b2* (*Figure 2—figure supplement 1B*).

We next examined if FXR binds to the enhancers that produce the identified eRNAs by analyzing published mouse liver ChIP-seq data for FXR (see Data availability) (*Lee et al., 2012*; *Thomas et al., 2010*). FXR binding was strongly enriched at the enhancers associated with FXR-induced eRNAs as were the enhancer marks H3K27ac and H3K4me1 (*Figure 2—figure supplement 1C*). The binding of FXR and the presence of histone marks at the *Fincor* enhancer region determined by ChIP-seq as compared with nascent transcripts detected by GRO-seq is shown in *Figure 2B*. These analyses support the conclusion that activation of FXR transcriptionally induces this series of eRNAs via chromatin binding at these enhancers, including *Fincor*.

We validated FXR binding at the enhancer region that produces *Fincor* using mouse liver ChIP (*Figure 2C*). We also examined binding of a well-known DNA binding partner of FXR, retinoid X receptor alpha (RXRα/NR2B1) (*Evans and Mangelsdorf, 2014*; *Zheng et al., 2018*), and bromodomain-containing protein 4 (BRD4), an acetylated histone reader protein that often binds at active enhancers (*Chen et al., 2016*; *Li et al., 2016*; *Rahnamoun et al., 2018*; *Sartorelli and Lauberth, 2020*) and a transcriptional coactivator of FXR (*Jung et al., 2020*). GW4064 treatment resulted in substantial increases in recruitment of both FXR and RXRα to the enhancer region close to the transcription start site of *Fincor* (arrow shown below in *Figure 2B*), whereas binding was not detected at a control region (*Figure 2C*). We also found that BRD4 occupancy was increased at this enhancer region after GW4064 treatment (*Figure 2C*). These results indicate that GW4064 activation of FXR leads to increased occupancy of the FXR/RXRα heterodimer and BRD4 to the enhancer region to upregulate *Fincor* eRNA in the liver.

We identified an inverted repeat1 (IR1) motif that is known to bind FXR (*Calkin and Tontonoz, 2012*; *Lee et al., 2006*) within the major FXR binding peak near the start site of *Fincor* (arrow shown below in *Figure 2B*), which we also refer to as FXRE (*Figure 2B and C*). We examined the functionality of this IR1 motif for mediating transcriptional activation by GW4064 using reporter assays (*Figure 2D*). We cloned the region containing the IR1 motif into the pGL4.23 luciferase reporter and generated a mutated IR1 motif construct as a comparison (*Figure 2D*). After transfection into human hepatic HepG2 cells, GW4064 treatment significantly elevated the luciferase activity of the reporter with the wild-type IR1 motif, but not with the mutated IR1 motif (*Figure 2D*). Together, these results suggest that GW4064-activated FXR directly upregulates *Fincor* expression.

## *Fincor* is a liver-specific nucleus-enriched eRNA

Because enhancers and eRNAs generally act in a tissue-specific manner (*Li et al., 2016*; *Sartorelli and Lauberth, 2020*), we examined the tissue-specific expression of *Fincor* in mice. Strikingly, *Fincor* is highly expressed in the liver and it is expressed at extremely low levels in most other tissues, except for a detectable, but still fairly low, level in the lung (*Figure 3A*). GW4064 treatment resulted in induction of *Fincor* specifically in the liver (*Figure 3A*). The level of *Fincor* detected in primary mouse hepatocytes (PMHs) isolated from GW4064-treated mouse liver was similar to that in the liver tissue from the same mouse, suggesting that the majority of *Fincor* is present in hepatocytes (*Figure 3B*).

By using the 5′ and 3′ rapid amplification of cDNA ends (RACE), we identified one transcript of *Fincor* that is approximately 3.7 kb in length (*Figure 3C*). However, based on RNA-seq, the length of *Fincor* is over 10 kb (*Figure 1C*), suggesting there are likely additional multiple RNA isoforms that we were not able to identify by RACE. We next analyzed the coding potential of *Fincor* utilizing a comparative genomic program, PhyloCSF. While adjacent genes *Gcnt1* and *Prune2* (*Figure 3D and E*) were correctly predicted to encode proteins, *Fincor* did not contain a potential protein-coding open reading frame (*Figure 3E*). Consistent with this bioinformatic prediction, a vector expressing *Fincor* failed to produce any proteins in an in vitro transcription/translation assay (*Figure 3F*), confirming that the *Fincor* transcript is an ncRNA. *Fincor* transcripts were enriched by binding to oligo-dT beads, suggesting that *Fincor* is 3′ polyadenylated (*Figure 3G*). Further, *Fincor* was detected in the nuclear compartment and GW4064 treatment increased the nuclear abundance of *Fincor* (*Figure 3H*), consistent with its potential transcriptional regulatory function. Together, these results from molecular biochemical characterization studies reveal that *Fincor* is a liver-enriched nuclear polyadenylated eRNA.

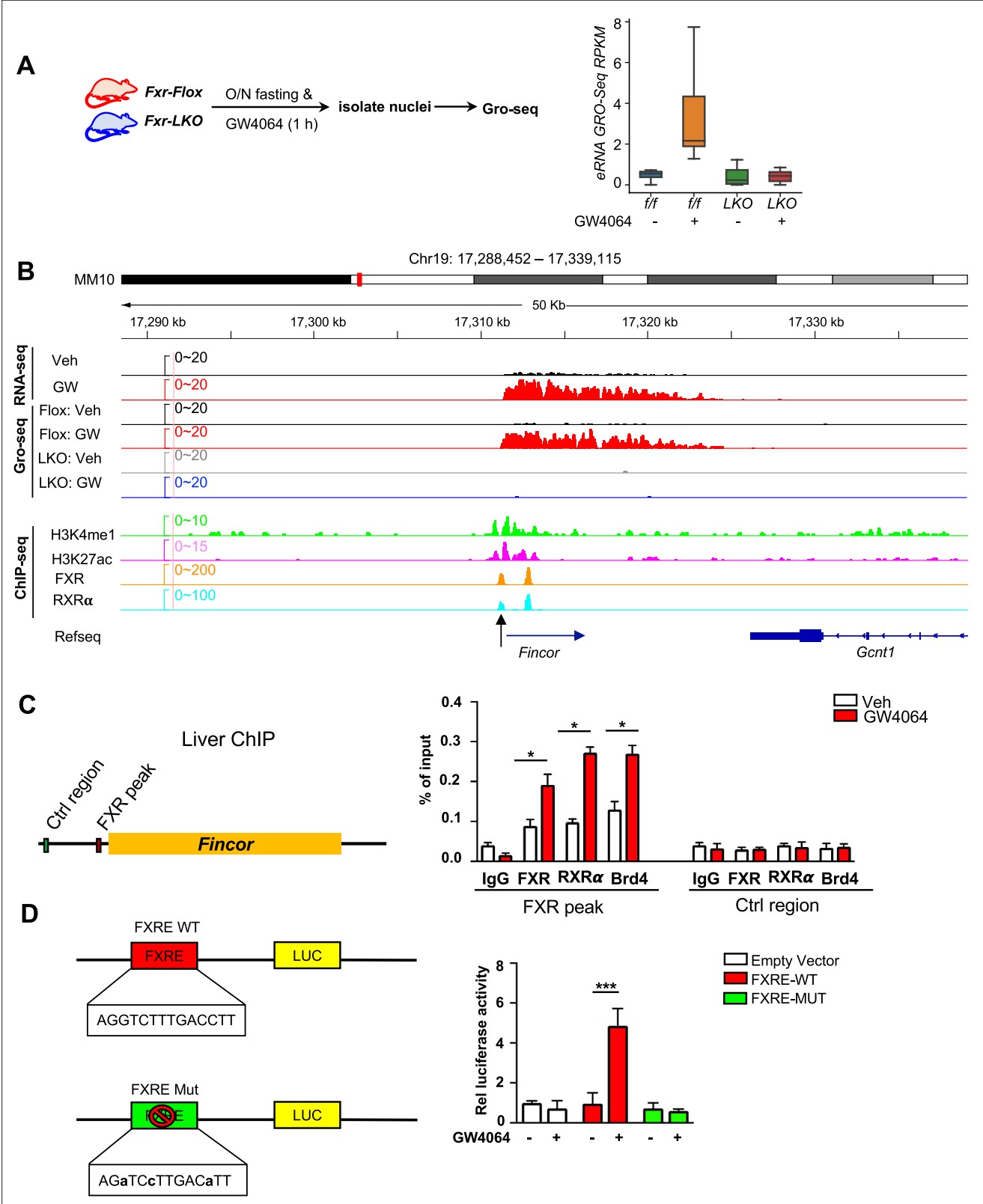

**Figure 2.** Ligand-activated farnesoid X receptor (FXR) directly activates transcription of enhancer RNAs including FXR-induced non-coding RNA (*Fincor*) in the liver. (**A**) Left: experimental outline. *Fxr* floxed (*Fxr-Flox*) and *Fxr* liver-specific knockout (*Fxr-LKO*) male mice were fasted overnight and treated with vehicle or GW4064, and livers were collected 1 hr later with nuclei isolated for global run-on sequencing (GRO-seq) (n=2/group). Right: a boxplot shows the GRO-seq signals for GW4064 upregulated eRNAs in different conditions. RPKM: reads per kbp per million. (**B**) IGV genome browser track showing

*Figure 2 continued on next page*

*Figure 2 continued*

RNA-seq, GRO-seq, and ChIP-seq signals in the *Fincor* locus. An arrow at the bottom points to the FXR ChIP-seq peak that contains an IR1 motif. (**C**) ChIP assays were performed in the same liver samples described in **Figure 1A** to detect FXR, retinoid X receptor alpha (RXRα), and bromodomain-containing protein 4 (BRD4) occupancy at the FXR binding peak region close to the transcription start site of *Fincor* (black arrow in **B**). (**D**) HepG2 cells were transfected with luciferase reporter expressing wild-type FXRE or mutant FXRE (see Materials and methods) for 24 hr before treatment with GW4064 for an additional 6 hr. Relative luciferase activities are shown. (**C–D**) Data are presented as mean ± SEM (n=3/group). Statistical significance was determined by the Student's t test with *p<0.05 and ***p<0.001.

The online version of this article includes the following figure supplement(s) for figure 2:

**Figure supplement 1.** Validation of Fincor expression and the analysis of the epigenetic features (presence of histone marks and binding of FXR and RXRα) at the enhancers that display FXR-induced eRNAs.

### *Fincor* is induced specifically by the hammerhead-type synthetic FXR agonists

To determine whether induction of the FXR-induced *Fincor* is ligand-specific, we examined the effects of several hammerhead-type synthetic FXR agonists, including GW4064, cilofexor, and tropifexor; a semi-synthetic agonist, OCA; and a non-hammerhead-type gut-specific agonist, fexaramine (**Figure 4A**; *Downes et al., 2003*; *Fang et al., 2015*). Remarkably, treatment with each of the hammerhead-type agonists for 1 hr resulted in a robust induction of *Fincor* (**Figure 4B**), whereas *Fincor* levels were unchanged after treatment with OCA or fexaramine for 1 hr (**Figure 4B**). Treatment with OCA for 4 hr or even 1 week treatment with OCA failed to induce hepatic *Fincor* in mice, while expression of *Nr0b2* was significantly induced (**Figure 4C and D**). Acute feeding with a diet supplemented with 0.5% cholic acid (CA), a primary BA, for 6 hr also failed to induce *Fincor* expression (**Figure 4E**). Collectively, these results demonstrate that *Fincor* is induced specifically by hammerhead-type FXR agonists.

### Generation of CRISPR/Cas9-mediated *Fincor*-LKD mice

To explore the functional role of hepatic *Fincor*, we utilized the CRISPR/Cas9 technique to generate *Fincor* liver-specific knockdown (*Fincor*-LKD) mice (**Figure 5—figure supplement 1A and B**). Adenoviral-mediated sgRNA expression in Cas9 mice resulted in downregulation of *Fincor* specifically in the liver by about 60%, but not in other tissues (**Figure 5B**). As FXR is a key regulator of BA, cholesterol, lipid, and glucose metabolism and *Fincor* is specifically regulated by FXR, *Fincor* may have a role in the metabolic process in physiology and disease. We examined liver triglyceride, cholesterol, BA, glycogen, and serum non-esterified fatty acids (NEFA) but *Fincor* downregulation did not result in any significant changes under physiological conditions (**Figure 5C**).

To explore the molecular signatures and pathways affected by *Fincor*, we examined global gene expression by RNA-seq analysis in mouse liver after *Fincor* knockdown (**Figure 5D**). While *Fincor* was markedly downregulated, the neighboring genes, such as *Gcnt1, Rfk, Pcsk5, and Prune2*, were largely unchanged (**Figure 5D**; **Figure 5—figure supplement 1C**). The RNA-seq analysis revealed 18 upregulated genes and 53 downregulated genes in *Fincor*-LKD liver (**Supplementary file 2**; **Figure 5E**). GO analysis of those downregulated genes indicated that these genes were enriched in pathways involved in fatty acid oxidation, organelle organization, and metabolic process (**Figure 5F**). Among the downregulated genes, *Ppp1r3g*, which has a role in controlling glycogen synthesis, and *Igfbp2*, which functions in insulin resistance, were markedly reduced (**Supplementary file 2**; **Figure 5—figure supplement 1D and E**). Among the upregulated genes, expression was substantially increased for *Eda2r*, which is a member of the tumor necrosis factor receptor superfamily and involved in inflammation, the immune response, and development (**Figure 5—figure supplement 1F**). *Fndc1*, which is involved in fibronectin matrix remodeling, was also suppressed by *Fincor* (and thus upregulated upon its knockdown, **Figure 5—figure supplement 1G**). These studies suggest that *Fincor* has a role in modulating metabolic homeostasis by regulating genes involved in metabolism and inflammation.

### Amelioration of hepatic steatosis mediated by tropifexor is independent of *Fincor* in diet-induced NASH mice

Tropifexor, also known as LJN452, is a highly potent hammerhead-type FXR agonist that is currently under clinical trials for NASH and PBC patients (*Kremoser, 2021*; *Sanyal et al., 2023*; *Tully et al., 2017*). Because *Fincor* is induced specifically by the hammerhead class of FXR agonist (**Figure 4**) and

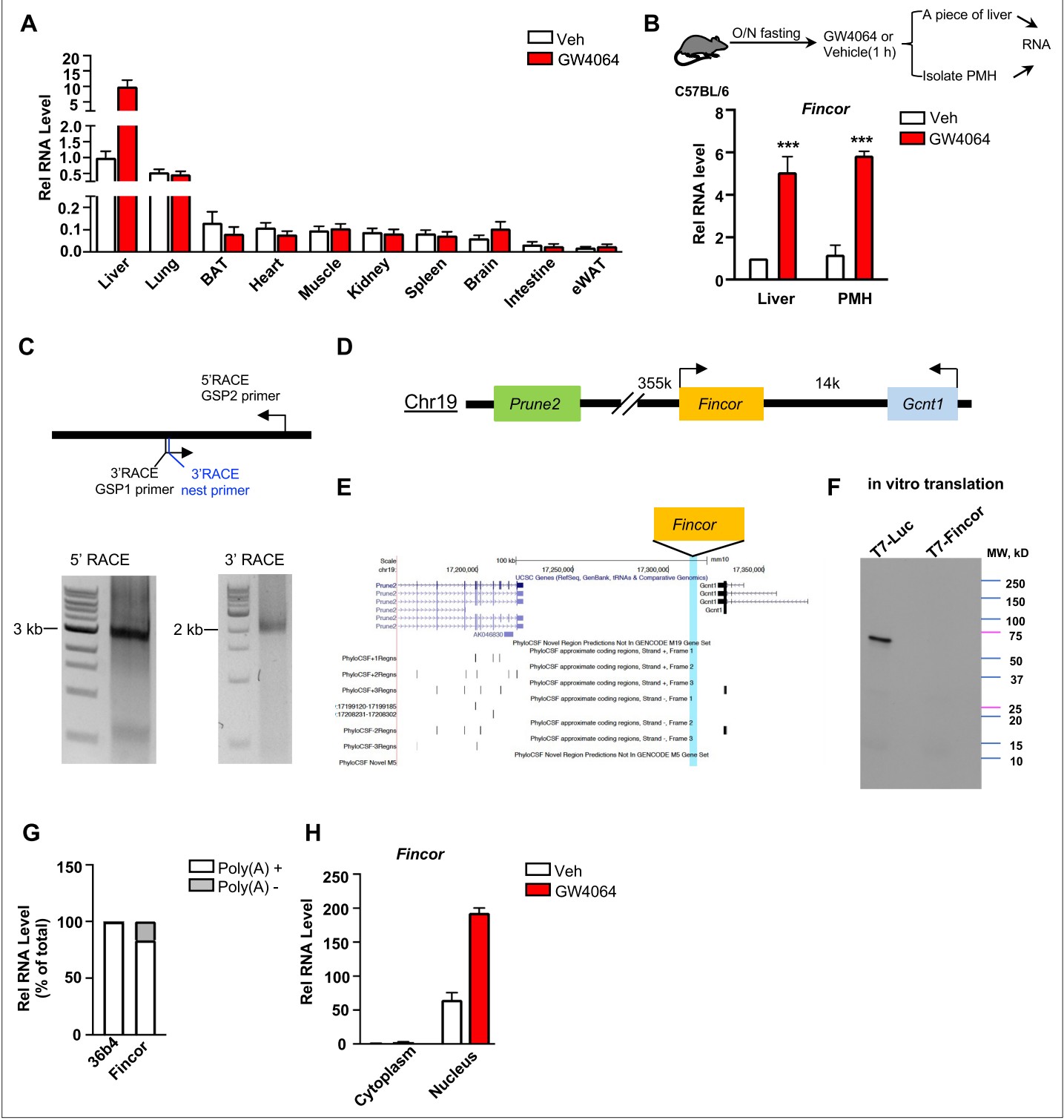

**Figure 3.** Farnesoid X receptor-induced non-coding RNA (*Fincor*) is a liver-specific nucleus-enriched eRNA. (**A**) Expression levels of *Fincor* in various tissues after GW4064 treatment. Data from C57BL/6 male mice fasted overnight and i.p. injected with vehicle or GW4064 (30 mg/kg) for 1 hr (n=2/group). Data are presented as mean ± SD. BAT, brown adipose tissue; eWAT, epididymal white adipose tissue. (**B**) C57BL/6 male mice were fasted overnight and injected i.p. with vehicle or GW4064 (30 mg/kg) for 1 hr. One small piece of liver was snap-frozen for later RNA isolation and the remaining part was used for immediate primary hepatocyte isolation. Then, RNAs were extracted from liver or primary hepatocytes and *Fincor* expression was measured (n=3/group). Data are presented as mean ± SEM. Statistical significance was determined by the Student's t-test with ***p<0.001. (**C**) Agarose gel electrophoresis of PCR products generated in 5' (left) and 3' (right) RACE of *Fincor* in liver samples. Primer locations are

*Figure 3 continued on next page*

*Figure 3 continued*

shown. RACE, rapid amplification of cDNA ends. GSP: gene-specific primer. (**D**) A schematic diagram showing location of *Fincor* relative to nearby genes in the mice genome. (**E**) PhyloCSF analysis of the coding potential of *Fincor*. (**F**) In vitro translation of *Fincor* using the Promega Transcend Non-Radioactive Translation Detection Systems. Luciferase is used as a control for coding RNA. (**G**) qPCR analysis of *Fincor*, *36b4* in Poly(A)+ and Poly(A)- RNA fractions from GW4064-treated mouse liver. (**H**) *Fincor* identified in the subcellular fractions using cellular fractionation assays. Primary hepatocytes were isolated from GW4064 or DMSO-treated mice and the cytoplasm and nucleus fractions of these hepatocytes were separated and both fractions were subjected to RNA extraction and qPCR (n=2/group). Data are presented as mean ± SD.

The online version of this article includes the following source data for figure 3:

**Source data 1.** Original file for the gel images shown in *Figure 3C*.

**Source data 2.** Original file for the gel images shown in *Figure 3C* with highlighted bands and sample labels.

**Source data 3.** Original file for the immunoblot image shown in *Figure 3F*.

**Source data 4.** Original file for the immunoblot image shown in *Figure 3F* with highlighted bands and sample labels.

has a potential role in the regulation of metabolism and inflammation (*Figure 5*), we hypothesized that *Fincor* may play a role in tropifexor-mediated beneficial effects on reducing NASH pathologies in mice.

We utilized a mouse model that had been fed the amylin liver NASH-promoting (AMLN) diet (*Hernandez et al., 2019*; *Sun et al., 2022*; *Zhao et al., 2018*), and examined the potential impact of liver-specific downregulation of *Fincor* on tropifexor's effects on NASH pathology. Cas9 mice fed the AMLN diet for 12 weeks were injected via tail veins with adenovirus expressing control sgRNA or *Fincor* sgRNA, respectively, and then treated daily with tropifexor (0.3 mg/kg) for 12 days (*Figure 6A*). In these mice, as a technical validation of RNA induction and knockdown, *Fincor* levels were significantly increased by FXR agonist tropifexor, and the increase was blocked by adenovirus expressing sgRNA for *Fincor* (*Figure 6B*).

We then examined the effect of tropifexor treatment and *Fincor* downregulation on hepatic steatosis in these mice. Tropifexor treatment markedly reduced neutral lipids determined by Oil Red O staining of liver sections (*Figure 6C*) and liver TG levels (*Figure 6D*), and these beneficial effects on reducing fatty liver were not altered by *Fincor* downregulation. Also, *Fincor* downregulation had little effect on liver cholesterol and gallbladder BA levels, although gallbladder BA levels were reduced by tropifexor (*Figure 6D*).

Consistent with the phenotypes, hepatic expression of key genes involved in BA synthesis was dramatically reduced by tropifexor treatment (i.e. *Cyp7a1 and Cyp8b1*), which is consistent with decreased gallbladder BA levels mediated by this agonist (*Figure 6E*). Similarly, tropifexor also lowered lipid synthesis genes (*Srebp1c, Lpin1, Scd1*), consistent with decreased liver TG levels. However, downregulation of *Fincor* did not result in changes in mRNA levels of these genes (*Figure 6E*). These results indicate that tropifexor-mediated beneficial effects on reducing hepatic steatosis are independent of *Fincor*.

### *Fincor* facilitates alleviation of liver inflammation by tropifexor in diet-induced NASH

Tropifexor ameliorated fibrotic NASH pathologies in preclinical studies (*Hernandez et al., 2019*; *Tully et al., 2017*) and has recently concluded phase 2 clinical trials for NASH patients (*Sanyal et al., 2023*). We, therefore, further examined the effects of *Fincor* downregulation on altering other NASH pathologies, including hepatocellular apoptosis, liver fibrosis, and inflammation.

In the same AMLN diet-fed mice as described above (*Figure 6*), analyses of liver sections revealed that tropifexor treatment reduced hepatocyte swelling/ballooning (hematoxylin and eosin [H&E] staining), decreased numbers of apoptotic cells (TUNEL staining), alleviated fibrosis (Sirius Red staining), and lowered infiltration of macrophages (F4/80 staining) (*Figure 7A*). Remarkably, these tropifexor-mediated beneficial effects on NASH pathologies were all markedly diminished by *Fincor* downregulation (*Figure 7A*). This was consistently found in various liver lobes and two representative pictures from two different lobes are shown (*Figure 7A*). In control experiments, *Fincor* downregulation in vehicle-treated mice did not result in marked changes in NASH pathologies (*Figure 7—figure supplement 1*). These results demonstrate that *Fincor* is required for tropifexor-mediated beneficial effects on reducing NASH pathologies, specifically reducing hepatic inflammation, fibrosis, and

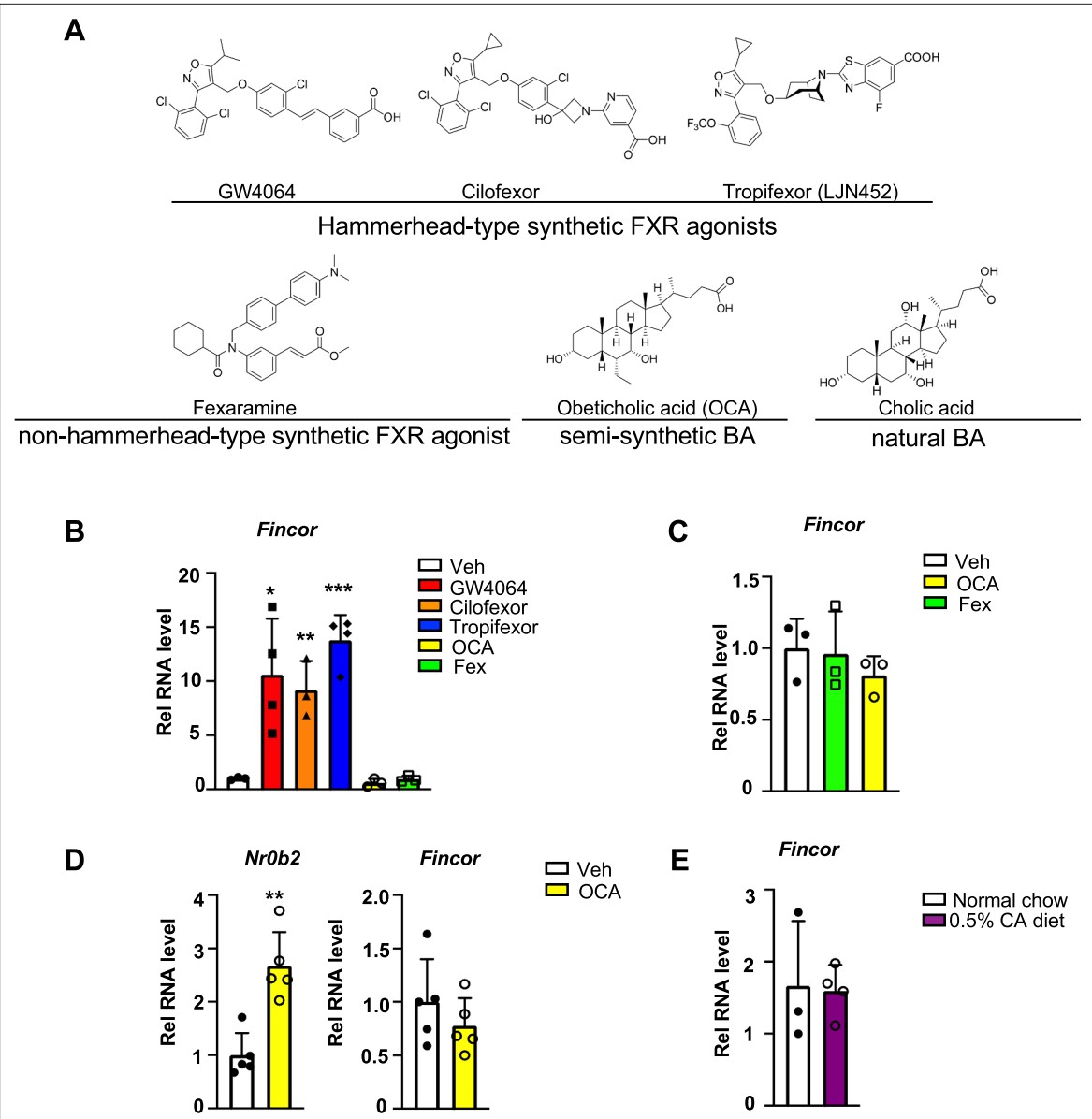

**Figure 4.** Farnesoid X receptor-induced non-coding RNA (*Fincor*) is induced by the hammerhead class of non-steroidal farnesoid X receptor (FXR) agonists, including GW4064 and tropifexor. (**A**) The chemical structures of the FXR agonists including the hammerhead class of synthetic FXR agonists, non-hammerhead-type synthetic agonist, semi-synthetic BA, and natural BA. BA: bile acid. (**B**) qPCR data showing *Fincor* expression levels in C57BL/6 mice liver respectively treated with GW4064 (30 mg/kg), cilofexor (30 mg/kg), tropifexor (0.5 mg/kg), fexaramine (100 mg/kg), or obeticholic acid (OCA) (20 mg/kg) for 1 hr (n=3–4/group). (**C**) *Fincor* expression levels in C57BL/6 mice liver treated with OCA (20 mg/kg) or fexaramine (100 mg/kg) for 4 hr (n=3/group). (**D**) Expression of *Fincor* in C57BL/6 mice liver after daily treatment with OCA (20 mg/kg) for 7 days (n=5/group). *Nr0b2* gene mRNA was measured as a positive control. (**E**) Expression of *Fincor* in C57BL/6 mice fed with 0.5% cholic acid (CA) diet for 6 hr (n=3–4/group). In panels B–E, all mice underwent overnight fasting before treatment. (**B–E**) Data are presented as mean ± SEM. Statistical significance was determined by the Student's t test with *p<0.05, **p<0.01, and ***p<0.001.

hepatocyte apoptosis. Consistent with these results, serum ALT and AST levels, indicators of liver damage, were significantly elevated after *Fincor* downregulation (*Figure 7B*). Protein levels of key inflammatory markers, IL1β and CCL2, in liver extracts were also elevated after *Fincor* downregulation (*Figure 7C*).

Consistent with the phenotypes from histological analyses, expression of hepatic genes involved in the above pathological process was altered by *Fincor* knockdown. For example, tropifexor treatment reduced mRNA levels of several genes that promote fibrosis (*Col1a1*, *Col1a2*, *Acta2*) and hepatic

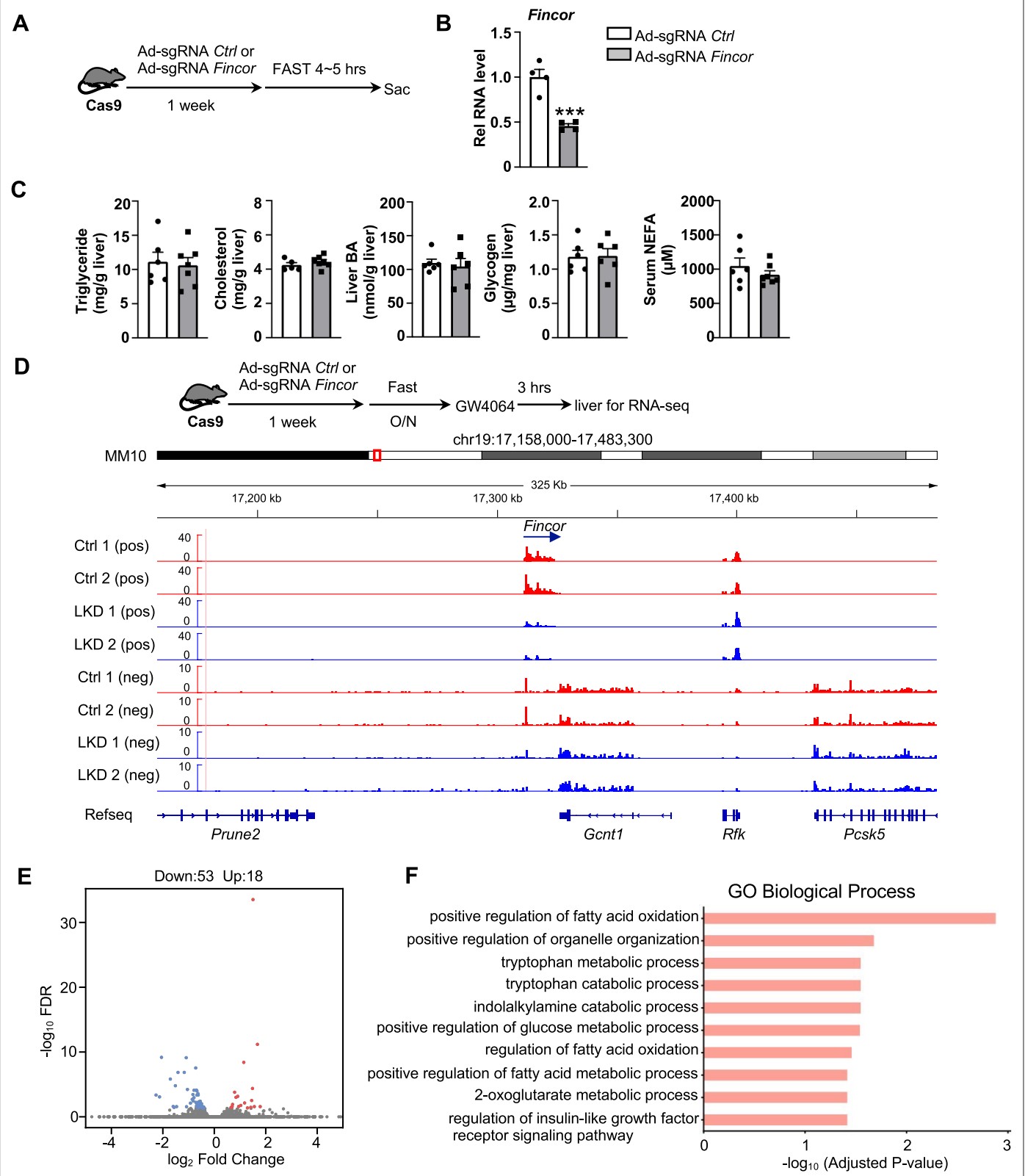

**Figure 5.** Generation of CRISPR/Cas9-mediated farnesoid X receptor-induced non-coding RNA (*Fincor*) liver-specific knockdown mice. (**A**) Experimental scheme: male Cas9 mice were infected with adenovirus expressing sgRNA for *Fincor* or a control for 1 week. Then the liver and serum were collected from these mice after 4–5 hr of fasting. (**B**) The expression of *Fincor* in the liver was measured by qPCR (n=4/group). (**C**) Hepatic triglyceride, cholesterol, bile acid, glycogen, and serum non-esterified fatty acids (NEFA) were measured (n=5–7/group). (**D**) Male Cas9 mice were infected with adenovirus

*Figure 5 continued on next page*

*Figure 5 continued*

expressing sgRNA for *Fincor* or control for 1 week. Then these mice were fasted overnight and treated with GW4064 for 3 hr before tissue collection. RNA-seq profiles of expression of hepatic *Fincor* and the adjacent genes were shown (n=2/group). (**E**) Genome-wide changes in mRNA expression shown in a volcano plot. The numbers refer to the number of genes up- or downregulated by twofold or more with an adjusted p-value <0.01. (**F**) Gene ontology analysis of biological pathways using DAVID Tools for genes downregulated after *Fincor* knockdown. (**B-C**) Data are presented as mean ± SEM. Statistical significance was determined by the Student's t test with ***p<0.001.

The online version of this article includes the following source data and figure supplement(s) for figure 5:

**Figure supplement 1.** Generation of farnesoid X receptor-induced non-coding RNA (Fincor) liver-specific knockdown mice by CRISPR-Cas9 method.

**Figure supplement 1—source data 1.** Original file for the gel image shown in *Figure 5—figure supplement 1*.

**Figure supplement 1—source data 2.** Original file for the gel image shown in *Figure 5—figure supplement 1* with highlighted bands and sample labels.

inflammation (*Eda2r, Ifng, Ccl3*), whereas these reductions were largely reversed by *Fincor* down-regulation (**Figure 7D**). We also detected increased expression of inflammatory genes (*Ccl2, Ccr2, Lcn2*) and an extracellular matrix remodeling gene (*Fndc1*) in liver with *Fincor* knockdown (**Figure 7D**). Tropifexor treatment suppressed hepatic apoptosis by reducing pro-apoptotic genes (*Ctsb, Ctss*) and upregulating anti-apoptotic genes such as *Bcl2* (**Warren et al., 2019**). Importantly, these effects were significantly reversed by *Fincor* downregulation (**Figure 7D**).

Collectively, these results demonstrate that in diet-induced NASH mice, pharmacological activation of FXR by tropifexor reduced fibrosis, apoptosis, and inflammation, which was dependent, at least in part, on the induction of *Fincor*.

### *Fincor* expression is increased in chronic liver disease with hepatic inflammation and liver injury

To determine whether expression of *Fincor* is altered in chronic liver disease, we utilized mouse models of NAFLD/NASH and cholestatic liver injury. Hepatic *Fincor* levels were significantly increased in mice fed with a high-fat diet (HFD) for 12 weeks (**Figure 8A**) and in mice fed an HFD with high fructose in drinking water for 12 weeks (**Figure 8B**). Elevated hepatic *Fincor* levels were also observed in mice treated with α-naphthylisothiocyanate (ANIT), a chemical inducer of liver cholestasis (**Jung et al., 2020**; **Kim et al., 2016**; **Figure 8C**), and in mice with bile duct ligation (BDL), a surgical method to induce cholestatic liver injury (**Figure 8D**).

The sequence of *Fincor* is moderately conserved between mice and humans as displayed in the UCSC genome browser (**Figure 8E**). Annotation in the NCBI genome data viewer of the human sequence region with similarity to mouse *Fincor* revealed an functionally uncharacterized human lncRNA, *XR_007061585.1*, in this region (**Figure 8F**). However, whether this sequence (or an as yet to be identified lncRNA) is functional or not has not been determined. To explore the potential changes or role of lncRNA *XR_007061585.1* in human liver pathological conditions, we measured hepatic levels of the transcripts in PBC and NAFLD patients. Compared to normal individuals, hepatic lncRNA *XR_007061585.1* levels were elevated in patients with PBC or NAFLD, but not in severe NASH-fibrosis patients (**Figure 8G and H**). These results demonstrate that hepatic levels of a potential human analog of *Fincor* are elevated in NAFLD and PBC patients, as *Fincor* is in mouse models of chronic liver disease with hepatic inflammation and liver injury. However, whether human lncRNA *XR_007061585.1* is analogous to mouse *Fincor* in terms of functions and mechanisms, and whether elevated *XR_007061585.1* levels may have a role in the disease progression or may be an adaptive response to liver injury remain to be determined.

### Discussion

FXR maintains metabolic homeostasis by transcriptional regulation of genes. Direct regulation of protein-coding genes by FXR, including *Nr0b2*, is well characterized. In this study we show that FXR also mediates its functions by induction of lncRNA genes, which vastly outnumber protein-coding genes. We further show that pharmacological activation of FXR by hammerhead-type agonists induces a liver-specific enhancer-derived lncRNA, which we named *Fincor*, that contributes to reduction of NASH pathologies in mice.

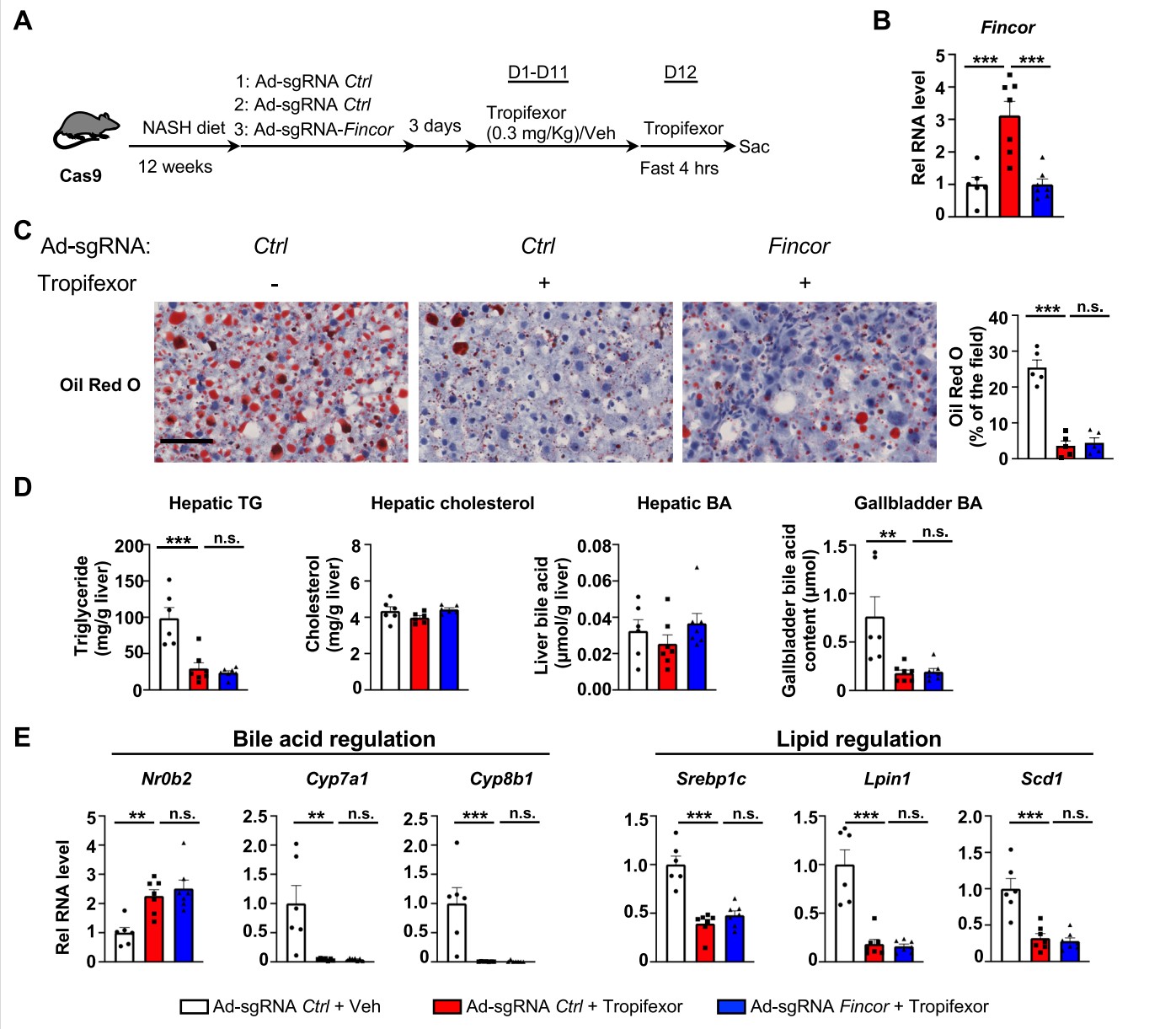

**Figure 6.** In diet-induced nonalcoholic steatohepatitis (NASH) mice, tropifexor-mediated beneficial effects on reducing hepatic steatosis are largely independent of farnesoid X receptor-induced non-coding RNA (*Fincor*). (**A–E**) Male Cas9 mice were fed a NASH diet for 12 weeks. The mice were randomly assigned to three groups and infected with adenovirus expressing sgRNA for *Fincor* or control. Three days later, the mice were daily treated with tropifexor (0.3 mg/kg) or vehicle from day 1 to day 11. On day 12, the mice were given the final treatment of tropifexor or vehicle and fasted for 4 hr before tissues were collected. (**A**) Experimental scheme. (**B**) Hepatic *Fincor* expression was measured (n=6–7/group). (**C**) Oil Red O staining of liver sections. Scale bar (50 μm). Image analyses were done using ImageJ and the areas of stained field were quantified (n=5/group). (**D**) Hepatic TG, hepatic cholesterol, gallbladder bile acid (BA), and hepatic BA levels were measured (n=6–7/group). (**E**) mRNA levels in the liver of the indicated genes involved in bile acid regulation and lipid regulation (n=6–7/group). (**B-E**) Data are presented as mean ± SEM. Statistical significance was determined by the one-way ANOVA (Sidak's multiple comparisons test) with *p<0.05, **p<0.01, and ***p<0.001. Ad, adenovirus; H&E, hematoxylin and eosin; TG, triglyceride; Veh, vehicle; ns, not significant.

*Fincor* is specifically induced by the hammerhead class of FXR agonists, such as GW4064, cilofexor, and tropifexor. GW4064 is the mother compound of these isoxazole-type hammerhead ligands but is not an ideal therapeutic agent because of its poor water solubility and pharmacokinetics (***Abel et al., 2010***). In contrast, tropifexor and cilofexor have better pharmacokinetics and are generally well tolerated in clinical trials for NASH and PBC patients (***Abenavoli et al., 2018***; ***Kremoser, 2021***; ***Sanyal et al., 2023***; ***Tully et al., 2017***), but the underlying mechanisms for their beneficial effects are poorly

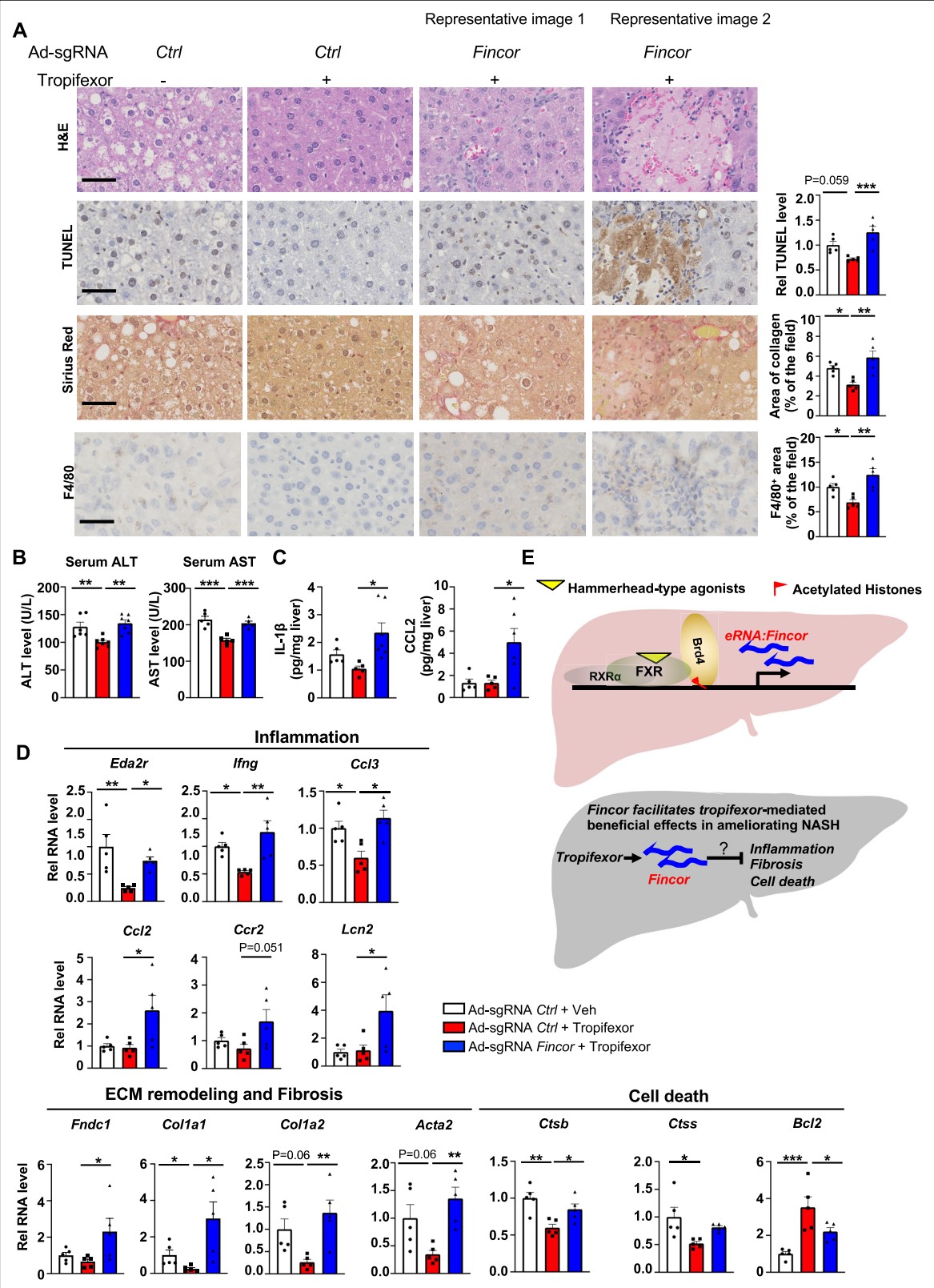

**Figure 7.** In diet-induced nonalcoholic steatohepatitis (NASH) mice, tropifexor-mediated beneficial effects on reducing liver fibrosis and inflammation are diminished by farnesoid X receptor-induced non-coding RNA (*Fincor*) downregulation. (**A**) Representative images from hematoxylin and eosin (H&E), TUNEL, Sirius Red, and F4/80 staining of liver sections from the same cohort of mice described in *Figure 6*. Scale bar (50 μm). Image analyses were done using ImageJ and the area of collagen staining, TUNEL, and F4/80 levels were quantified (n=5/group). (**B**) Serum ALT and AST levels

**Figure 7 continued**

were measured (n=5/group). (**C**) IL-1β and CCL2 levels in the liver tissues were determined by ELISA (n=5/group). (**D**) mRNA levels in the liver of the indicated genes involved in inflammation, fibrosis, and cell death (n=5/group). (**E**) Model: *Fincor* is a liver-enriched eRNA that is induced specifically by hammerhead-type farnesoid X receptor (FXR) agonists (top). In diet-induced NASH mice, *Fincor* is required for tropifexor-mediated beneficial effects on reducing hepatic inflammation, fibrosis, and cell death with the mechanisms to be determined (bottom). (**A–D**) Data are presented as mean ± SEM. Statistical significance was determined by the one-way ANOVA (Sidak's multiple comparisons test) with *$p<0.05$, **$p<0.01$, and ***$p<0.001$.

The online version of this article includes the following figure supplement(s) for figure 7:

**Figure supplement 1.** The effects of farnesoid X receptor-induced non-coding RNA (*Fincor*) downregulation on nonalcoholic steatohepatitis (NASH) pathologies.

understood. Intriguingly, a recent study showed that the gene signature regulated by tropifexor-activated FXR appears to be broader than that of OCA, partly because the tropifexor backbone allows a more favorable interaction of FXR with coactivators or epigenomic modulators (*Hernandez et al., 2019*). Further, tropifexor was shown to regulate distinct sets of genes in experimental NASH as compared to other FXR agonists, particularly genes involved in fibrosis, inflammation, and oxidative stress (*Hernandez et al., 2019*). Utilizing CRISPR/Cas9-mediated liver-specific knockdown of *Fincor* in diet-induced NASH mice, we demonstrate that beneficial effects on reducing liver fibrosis, inflammation, and apoptosis mediated by tropifexor were largely dependent on *Fincor* (Model, *Figure 7E*).

While this study focused on regulation of *Fincor* by pharmacological activation of FXR, physiological and pathological regulations of *Fincor* appear to be complex. Although *Fincor* can be induced by the hammerhead class of FXR agonists, it was not induced by the endogenous FXR ligand, CA. This implies that *Fincor* may not contribute to the physiological functions of FXR. In an effort to investigate the potential role of *Fincor* in the pathological conditions, we observed that *Fincor* levels were elevated in mouse models of cholestasis and NASH as well as in human PBC and NAFLD patients, where BA metabolism is dysregulated. Since different BAs can activate or repress the gene-regulating function of FXR (*Wahlström et al., 2016*), altered BA composition in these pathological conditions may contribute to induction of *Fincor*. Further, binding peaks for multiple nuclear receptors, FXR, LXR, PPARα, RXRα, and HNF-4α, were detected in the *Fincor* locus so that regulation of *Fincor* likely involves the combinatorial regulation by multiple nuclear receptors (*Supplementary file 3a*). Interestingly, occupancy of the nuclear receptor PPARα at the enhancer region was increased in fasted mice (*Supplementary file 3b*). It will be interesting to investigate whether and how *Fincor* is differently regulated by these nuclear receptors in response to physiological and pathological cues.

The roles of regulatory RNAs in liver function and diseases and their potentials as therapeutic targets are increasingly being appreciated (*Brocker et al., 2020*; *Li et al., 2021*; *Sallam et al., 2016*; *Zhao et al., 2018*). For example, a critical role for an oxysterol nuclear receptor LXR-induced lncRNA, *LeXis*, in feedback modulation of cholesterol biosynthesis has been shown (*Sallam et al., 2016*). Recently, the role of an lncRNA, *Pair*, in liver phenylalanine metabolism has been demonstrated (*Li et al., 2021*). Enhancer RNAs are a less-characterized class of lncRNAs and are highly associated with enhancer functions in gene regulation (*Li et al., 2016*). Numerous studies have revealed transcriptional roles for eRNAs in various cellular processes (*Lai and Shiekhattar, 2014*; *Lam et al., 2014*; *Li et al., 2016*; *Sartorelli and Lauberth, 2020*) but most previous eRNA studies have used cultured cells (*Hsieh et al., 2014*; *Li et al., 2013*), with only a few in vivo studies in mouse models (*Mirtschink et al., 2019*; *Tang et al., 2023*). In our current study, through integrative analysis of transcriptome and histone mark ChIP-seq, we identified a group of FXR-regulated eRNAs, including the highly induced *Fincor*. Our current work characterized the role of *Fincor* in gene regulation and in mediating beneficial pharmacological effects of tropifexor in NASH, representing important progress in understanding the roles of eRNAs in vivo. Future work is warranted to elucidate the exact mechanisms by which *Fincor* facilitates action of FXR agonists to alleviate inflammation, fibrosis, and apoptosis.

RNA inside the cells usually associates with different RNA binding proteins (RBPs) (*Gerstberger et al., 2014*). We identified potential binding proteins of *Fincor* using the ATtRACT database (*Giudice et al., 2016*). The top four candidates for *Fincor* binding are KHDRBS1, RBM38, YBX2, and YBX3 (*Supplementary file 4*). KHDRBS1 and RBM38 have been reported to have important roles in RNA processing (*Bielli et al., 2011*; *Zou et al., 2021*). YBX2 and YBX3 belong to the Y-box (YBX) protein family, which have been linked to diverse forms of RNA metabolism and many other processes, including cell proliferation, DNA repair, stress responses, development, and inflammation (*Kleene,*

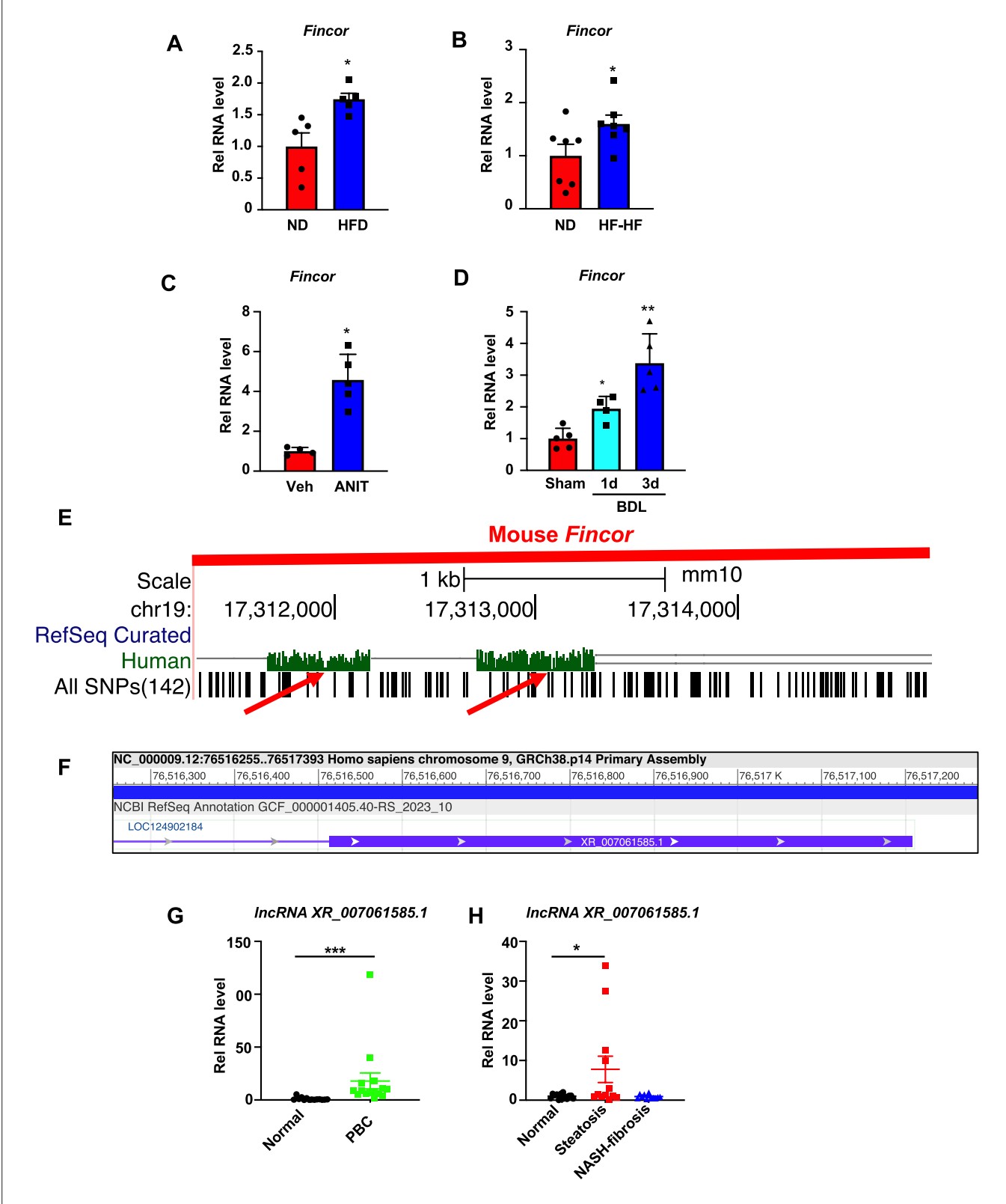

**Figure 8.** Hepatic expression of farnesoid X receptor-induced non-coding RNA (*Fincor*) is elevated in liver disease associated with inflammation and fibrosis. (**A**) C57BL/6 mice were fed with a high-fat diet for 12 weeks. The liver RNAs were extracted and *Fincor* expression was measured (n=5/group). (**B**) C57BL/6 mice were fed with a high-fat diet with high fructose water for 12 weeks. The liver RNAs were extracted and *Fincor* expression was measured (n=7/group). (**C**) C57BL/6 mice were treated with α-naphthylisothiocyanate (ANIT) (75 mg/kg) for 48 hr and then sacrificed after 5 hr of fasting. The liver

*Figure 8 continued on next page*

*Figure 8 continued*

RNAs were extracted, and *Fincor* expression was measured (n=4–5/group). (**D**) C57BL/6 mice were bile duct ligated for 1 day or 3 days. They were then sacrificed after 5 hr of fasting. The liver RNAs were extracted and *Fincor* expression was measured (n=4–5/group). (**E**) *Fincor* conservation between mice and human as displayed in the UCSC Genome Browser. Red arrows indicate the conserved region. (**F**) Human long non-coding RNA (lncRNA) *XR_007061585.1* with sequence similarity to mouse *Fincor* annotated in the NCBI genome data viewer. (**G**) Expression of lncRNA *XR_007061585.1* in liver samples from normal individuals or patients with primary biliary cholangitis (PBC) (n=14–15/group). (**H**) Expression of lncRNA *XR_007061585.1* in liver samples from normal individuals or patients with nonalcoholic fatty liver disease (NAFLD)-associated steatosis (n=12–15/group). (**A–D, G–H**) Data are presented as mean ± SEM. Statistical significance was determined by the Student's t test with *p<0.05, **p<0.01, and ***p<0.001.

*2018*). Whether these predicted RBPs interact with *Fincor* and how they contribute to phenotypes should be investigated in future experimentation to understand the mechanisms involved in *Fincor*-regulated hepatocyte function.

There is limitation of some of our approaches that cannot fully dissect the underlying mechanisms of *Fincor*. Currently, the function of eRNA loci can be attributed to a functional eRNA transcript, or binding of transcription factors to this region, or the transcription process itself, or some combination of these effects (*Li et al., 2016*; *Sartorelli and Lauberth, 2020*). In our studies to decrease expression of *Fincor*, the region containing the FXR binding site was deleted (*Figure 5—figure supplement 1A*, right panel). While the expression of *Fincor* transcripts was significantly reduced, we cannot rule out whether decreased binding of FXR or decreased transcription contribute to the observed changes in phenotype. To directly investigate the function of *Fincor* transcripts, downregulation of the transcript with antisense oligonucleotides together with overexpression of *Fincor* in the liver will be required in future experiments.

FXR is increasingly recognized as an important therapeutic target for enterohepatic diseases, but the development of clinically applicable and more targeted FXR-based therapy is still challenging. In this study, we provide the first characterization of an eRNA, *Fincor*, induced by pharmacological activation of FXR and show that *Fincor* has a beneficial role in reducing liver fibrosis and inflammation in dietary NASH mice. Complete understanding of the function and mechanisms of *Fincor* may provide novel insights for the development of desirable therapy for NASH and other chronic liver diseases.

## Materials and methods

### Animal experiments

All animal studies were performed according to procedures approved by the Institutional Animal Care & Use Committee at the University of Illinois at Urbana-Champaign (protocol # 17009) and were in accordance with National Institutes of Health guidelines. Mice were maintained in 12/12 hr light/dark cycles and fed standard rodent chow. *Fxr-LKO* mice were generated by breeding *Fxr-Flox* mice with Albumin-Cre mice (The Jackson Lab). *Fincor*-LKD mice were generated as previously reported (*Zhao et al., 2018*). Briefly, Cas9 transgenic mice (JAX #024858) were injected via the tail vein with adenoviruses (approximately $5 \times 10^8$ PFU) expressing two sgRNAs targeting *Fincor* (sgRNA1: *GGGT TAAGAGCTGTAGGCTG* and sgRNA2: ACTTCTATGTCCAACAACCG). The sequences of sgRNAs were designed using a CRISPR design tool (http://crispr.mit.edu/).

Mice were given a single dose of vehicle or 30 mg/kg GW4064 (in corn oil, Tocris Bioscience, #2473) after overnight fasting. Mice were treated with 0.5 mg/kg tropifexor (in corn oil, MedChem Express, HY-107418), 30 mg/kg cilofexor (in corn oil, MedChemExpress, HY-109083), 100 mg/kg fexaramine (in 0.5% methylcellulose, MedChem Express, HY-10912), and 20 mg/kg OCA (in 0.5% methylcellulose, MedChem Express, HY-12222) as indicated. C57BL6 mice were fed with a chow diet containing 0.5% CA for 6 hr (*Jung et al., 2020*).

To induce cholestasis, mice were treated by gavage with 75 mg/kg ANIT for 48 hr as previously reported (*Jung et al., 2020*; *Kim et al., 2016*; *Kim et al., 2020*). Cholestasis was also induced by BDL or sham operation in mice for 24 hr or 72 hr (*Li et al., 2018*).

To induce dietary obesity, mice were fed an HFD (TD88137; Harlan Teklad) or an HFD with 25% fructose in water (high fat/high fructose) for 12 weeks (*Seok et al., 2021*).

To investigate the effect of liver-specific downregulation of *Fincor* on NASH, male Cas9 mice were fed the AMLN diet (Research Diets, D09100310, 40 kcal% fat, 2% cholesterol, 20 kcal% fructose) for 12 weeks and then, were injected with adenovirus expressing control sgRNA or sgRNA targeting

*Fincor.* Administration of tropifexor was started 3 days later and given at 0.3 mg/kg dissolved in corn oil.

For all the mice experiments, the mice were randomly assigned to control group or treatment groups as needed.

## RNA-seq

C57BL/6 mice were fasted overnight and i.p. injected with vehicle or GW4064 (30 mg/kg) for 1 hr, and livers were collected (n=4 mice for either vehicle or treated group). Total RNA from each liver was extracted by RNeasy kit (QIAGEN), and two randomly selected mice liver RNAs were pooled for RNA-seq (two RNA-seq reactions from four mice livers for either vehicle or treated group). RNA-seq was performed as previously described (*Byun et al., 2018*; *Byun et al., 2020*; *Seok et al., 2018*). Ribosomal RNA was removed with the Ribozero HMR Gold kit (Illumina). The sequencing library was generated by the following methods described below.

## Construction of strand-specific RNA-seq libraries

Construction of the RNA-seq libraries and sequencing on the Illumina NovaSeq 6000 were performed at the Roy J. Carver Biotechnology Center at the University of Illinois at Urbana-Champaign. After DNase digestion, purified total RNAs were analyzed on a Fragment Analyzer (Agilent) to evaluate RNA integrity. The total RNAs were converted into individually barcoded polyadenylated mRNA-seq libraries with the Kapa HyperPrep mRNA kit (Roche). Libraries were barcoded with Unique Dual Indexes which have been developed to prevent index switching. The adaptor-ligated double-stranded cDNAs were amplified by PCR for eight cycles with the Kapa HiFi polymerase (Roche). The final libraries were quantitated with Qubit (Thermo Fisher) and the average cDNA fragment sizes were determined on a Fragment Analyzer. The libraries were diluted to 10 nM and further quantitated by qPCR on a CFX Connect Real-Time qPCR system (Bio-Rad) for accurate pooling of barcoded libraries and maximization of number of clusters in the flowcell.

## Sequencing of libraries in the NovaSeq

The barcoded RNA-seq libraries were loaded on one SP lane on a NovaSeq 6000 for cluster formation and sequencing. The libraries were sequenced from one end of the fragments for a total of 100 bp. The fastq read files were generated and demultiplexed with the bcl2fastq v2.20 Conversion Software (Illumina, San Diego, CA, USA). The quality of the demultiplexed fastq files was evaluated with the FastQC software, which generates reports with the quality scores, base composition, k-mer, GC and N contents, sequence duplication levels, and overrepresented sequences.

## GRO-seq

To harvest the nuclei from mouse liver cells, the liver was harvested at indicated time and washed with a cold swelling buffer (10 mM Tris-HCl, pH 7.5, 2 mM $MgCl_2$, 3 mM $CaCl_2$, 2 U/ml Superase-In). The nuclei were prepared by Dounce homogenization in cold swelling buffer and filtered using a cell strainer (100 µm, BD Biosciences). Nuclei were collected by centrifugation at 400 × $g$ for 10 min, then resuspended in the lysis buffer (swelling buffer with 10% glycerol and 1% IGEPAL) and incubated on ice for 5 min. Nuclei were washed twice with the lysis buffer and resuspended at a concentration of $10^8$ nuclei/ml in the freezing buffer (50 mM Tris-HCl, pH 8.3, 40% glycerol, 5 mM $MgCl_2$, 0.1 mM EDTA). We then followed our previous method (*Li et al., 2013*; *Oh et al., 2021*) to conduct nuclear run-on and GRO-seq library preparation. Briefly, the nuclei in freezing buffer were subjected to the nuclear run-on reaction by mixing with an equal volume of run-on buffer (10 mM Tris-HCl, pH 8.0, 5 mM $MgCl_2$, 1 mM dithiothreitol, 300 mM KCl, 20 units of Superase-In, 1% sarkosyl, 500 µM ATP, GTP, Br-UTP, and 2 µM CTP) for 5 min at 30°C. The BrU-labeled run-on RNAs were extracted by TRIzol and purified by anti-BrdU agarose beads (Santa Cruz Biotech, sc-32323 AC). The run-on RNAs were then subjected to end repair by T4 PNK, poly-adenylation, and then to cDNA first strand synthesis by a custom primer (oNTI223) that allows circularization of the cDNA. This cDNA then was re-linearized by Ape1 (NEB), size selected by TBE gel electrophoresis, and the products of the desired size were excised (~320–350 bp) for final library prep and sequencing. GRO-seq samples were run on a NextSeq 500 sequencer from Illumina with a single-end 80 nt model.

## Histological analyses

For histology, tissues were dissected and immediately fixed in 10% formalin overnight and processed for paraffin embedding and H&E staining. Paraffin-embedded liver sections were incubated with F4/80 antibody, and antibody was detected using a peroxidase-based method (Abcam, ab64238). Liver collagen was detected by Sirius Red staining (Abcam, ab246832) and apoptosis was detected by TUNEL staining (Millipore, S7100). For Oil Red O staining, liver tissue was frozen in OCT compound (Sakura Finetek, 4583), sectioned, and stained. Liver sections were imaged with a NanoZoomer Scanner (Hamamatsu) and quantification was done using NIH ImageJ.

## Metabolic analyses

Hepatic levels of TG (Sigma, MAK266), cholesterol (Sigma, MAK043), glycogen (Biovision, K646-100), total BA levels (Diazyme, DZ042A), serum NEFA (Sigma, MAK044), serum ALT (Sigma, MAK052), and serum AST (Sigma, MAK055) were determined according to the manufacturer's instructions. Mouse liver IL-1β (R&D Systems, MLB00C) and CCL2 (R&D Systems, DY479-05) were detected by commercially available ELISA kit.

## Liver samples of PBC and NAFLD patients

Liver specimens from normal organ donors, and patients with PBC or NAFLD were obtained from the Liver Tissue Procurement and Distribution System. The samples were unidentifiable, and thus, ethical approval was not required. Hepatic *XR_007061585.1* levels were measured by qPCR.

## Mouse liver ChIP

Liver ChIP assay was performed as described previously (*Jung et al., 2020*; *Seok et al., 2018*). Briefly, chromatin extracts were prepared from *Fxr-Flox* and *Fxr-LKO* mouse livers after treatment of the mice with GW4064 which was followed by preclearing and immunoprecipitation using control IgG or FXR antibody (Novus Biologicals, NBP2-16550; Santa Cruz, sc-25309), RXRα antibody (Proteintech, catalog no. 21218-1-AP), or BRD4 antibody (Bethyl Laboratories, catalog #A301-985A50). Enrichment in chromatin precipitates of gene sequences was measured by qPCR using primers listed (*Supplementary file 5*).

## PMHs and HepG2 cells

Primary hepatocytes were isolated from C57BL/6 mice by collagenase (Worthington Biochemical Corporation, LS004188) perfusion and maintained in William's E Medium with primary hepatocyte maintenance supplements (Gibco #CM4000) in six-well plates as described previously (*Jung et al., 2020*; *Kim et al., 2015*). HepG2 cells were purchased from the American Type Culture Collection and grown in Dulbecco's modified eagle medium containing 10% FBS and 1% penicillin and streptomycin. This cell line was not authenticated after purchase but routinely tested negative for mycoplasma contamination.

## Luciferase reporter assay

The enhancer region containing the FXR binding element was amplified by PCR and cloned into the pGL4.23 vector (Promega). HepG2 cells were transiently transfected with FXR (100 ng/well) and RXRα (5 ng/well) in combination with reporters containing the wild-type FXRE or mutated FXRE (200 ng/well) (*Supplementary file 5*). β-Galactosidase plasmid (200 ng/well) was also transfected as an internal control. Cells were treated with GW4064 or DMSO after transient transfection. Six hours later, the cells were harvested. All reporter assays were repeated at least three times in triplicates.

## RACE

5' and 3' RACE assays were performed using a SMARTer RACE kit (Clontech) according to the manufacturer's instructions. The resulting PCR products were separated by electrophoresis in agarose gels and cloned into the pRACE vector provided by the kit. The transcription start sites and end sites of *Fincor* were determined by sequencing. The gene-specific primers used for 5' and 3' RACE are listed in *Supplementary file 5*.

## In vitro transcription and translation

Expression plasmids for luciferase and *Fincor* were mixed with a Coupled Reticulocyte Lysate System (Promega). After incubating at 30°C for 60 min, translated products were separated on 4–20% gradient SDS polyacrylamide gels and transferred to PVDF membranes. Chemiluminescent detection of in vitro translated protein was performed following the manufacturer's protocol (Promega).

## RT-qPCR

Total RNA was extracted using the RNeasy Mini Kit (QIAGEN, 74104) and 2 µg of RNA was reverse-transcribed and RNA expression was normalized relative to that of 36B4. The qPCR primers are shown in **Supplementary file 5**.

## Subcellular fractionation

Using a Cytoplasmic and Nuclear RNA Purification Kit (Norgen, Thorold, ON, Canada), the cytoplasm and nucleus fractions from primary hepatocytes isolated from livers of mice treated with GW4064 or DMSO were separated, and both fractions were subjected to RNA extraction and qRT-PCR.

## Immunoblotting analysis

Total liver lysates were prepared as described before (**Sun et al., 2022**). Antibodies for immunoblotting for β-ACTIN (#4970) were purchased from Cell Signaling Technology. Antibodies for immunoblotting for FXR (sc-25309) were purchased from Santa Cruz.

## *Fincor* polyadenylation study

Total RNA was prepared using the RNeasy mini prep kit (QIAGEN). Poly(A)+ and poly(A)- RNA was separated using a Dynabeads mRNA Purification Kit (Thermo Fisher Scientific) according to the manufacturer's instructions. Briefly, total RNA was incubated with the Dynabeads/binding buffer suspension at room temperature for 5 min and the reaction tubes were placed on a magnet until solution was clear. The supernatant containing poly(A)- RNA was saved. The beads with poly(A)+ RNA were washed three times with washing buffer provided by the kit. RNA was extracted from the supernatant and beads respectively using TRIzol. qPCR was performed to analyze levels of *Fincor* and 36b4 in the poly(A)+ and poly(A)- RNA fractions.

## Genomics analysis

RNA-seq and GRO-seq reads were mapped to the mouse reference genome mm10 with STAR aligner (**Dobin et al., 2013**). Transcript quantifications were done with the HOMER tool set (**Heinz et al., 2010**) and enhancer RNAs were identified based on H3K27ac ChIP-seq in mouse liver. Briefly, intergenic H3K27ac ChIP-seq peaks were selected as putative enhancer regions in mouse liver. Then ±3 kb regions around putative enhancers with RNA-seq signal (>1 RPKM [reads per kbp per million]) were considered as putative eRNAs in mouse livers. Overlapped eRNA regions were merged, and redundant ones were removed. In addition, any ±3 kb extended eRNA regions that overlapped with protein-coding genes were further removed to avoid transcriptional readthrough from genes. We used DESeq2 (**Love et al., 2014**) to identify significantly regulated eRNAs with a cutoff of (FDR <0.05, $\log_2$FC >1). Public ChIP-seq datasets were obtained from ENCODE or GEO (see Data availability).

## Statistical analysis

Statistical analysis was performed using GraphPad Prism 9. Statistical differences were evaluated using the two-tailed unpaired Student's t-test for comparisons between two groups, or ANOVA and appropriate post hoc analyses for comparisons of more than two groups. Statistical methods and corresponding p-values for data shown in each panel are indicated in figure legends.

## Acknowledgements

We thank Deepa Prakashini Govindasamy Rajagopal for the help with tissue genotyping. We thank Tiangang Li at University of Oklahoma for providing bile duct-ligated mouse liver samples, and Kristina Schoonjans at Ecole Polytech, Switzerland, for providing *Fxr-Flox* mice. We also thank the Liver Tissue Cell Distribution System, University of Minnesota (NIH Contract # HHSN276201200017C), for

providing liver specimens of NAFLD and PBC patients and individuals without liver disease. This study was supported by an American Diabetes Association Postdoctoral Fellowship to JC (1-19-PDF-117), and by a John and Rebekah Harper Fellowship to RW. WL is a Cancer Prevention and Research Institute of Texas (CPRIT) Scholar (RR160083). This work is supported by funding from NIH (K22CA204468 and R01GM136922), Welch Foundation (AU-2000-20220331) and John S Dunn foundation to WL, and grants from the NIH (R01 DK062777 and R01 DK095842) to JKK.

## Additional information

### Funding

| Funder | Grant reference number | Author |
| --- | --- | --- |
| American Diabetes Association | Postdoctoral fellowship 1-19-PDF-117 | Jinjing Chen |
| John and Rebekah Harper | Graduate student fellowship | Ruoyu Wang |
| Cancer Prevention and Research Institute of Texas | RR 160083 | Wenbo Li |
| Welch Foundation | AU-2000-20220331 | Wenbo Li |
| National Institutes of Health | R01GM136922 | Wenbo Li |
| National Institutes of Health | K22CA204468 | Wenbo Li |
| National Institutes of Health | R01 DK062777 | Jongsook Kemper |
| National Institutes of Health | R01 DK095842 | Jongsook Kemper |
| John S. Dunn Foundation | Collaborative research award 2019 | Wenbo Li |

The funders had no role in study design, data collection and interpretation, or the decision to submit the work for publication.

### Author contributions

Jinjing Chen, Conceptualization, Data curation, Formal analysis, Funding acquisition, Validation, Investigation, Methodology, Writing – original draft, Project administration, Writing – review and editing; Ruoyu Wang, Data curation, Software, Formal analysis, Funding acquisition, Investigation, Methodology, Writing – original draft, Writing – review and editing; Feng Xiong, Data curation, Investigation, Methodology; Hao Sun, Data curation, Formal analysis, Investigation; Byron Kemper, Formal analysis, Investigation, Methodology, Writing – original draft, Writing – review and editing; Wenbo Li, Jongsook Kemper, Conceptualization, Resources, Formal analysis, Supervision, Funding acquisition, Investigation, Methodology, Writing – original draft, Project administration, Writing – review and editing

### Author ORCIDs

Jinjing Chen http://orcid.org/0000-0002-0612-8553
Ruoyu Wang http://orcid.org/0000-0002-3644-1284
Wenbo Li http://orcid.org/0000-0002-9042-5664
Jongsook Kemper https://orcid.org/0000-0002-5534-0286

### Ethics

Liver specimens from normal organ donors, and patients with PBC or NAFLD were obtained from the Liver Tissue Procurement and Distribution System (NIH Contract # HHSN276201200017C). The samples were unidentifiable, and thus, ethical approval was not required.

All animal studies were performed according to procedures approved by the Institutional Animal Care & Use Committee at the University of Illinois at Urbana-Champaign (protocol # 17009) and were in accordance with National Institutes of Health guidelines.

Reviewer #1 (Public review): https://doi.org/10.7554/eLife.91438.3.sa1
Reviewer #2 (Public Review): https://doi.org/10.7554/eLife.91438.3.sa2
Author response https://doi.org/10.7554/eLife.91438.3.sa3

---

# Additional files

## Supplementary files

• Supplementary file 1. Sequencing data generated in this study and a list of Farnesoid X receptor (FXR)-associated eRNAs.

• Supplementary file 2. Differentially expressed genes in the FXR-induced non-coding RNA (*Fincor*)-downregulated mice liver.

• Supplementary file 3. The binding of transcription factors at the FXR-induced non-coding RNA (*Fincor*) locus.

• Supplementary file 4. Predicted RNA binding proteins (RBPs) binding to FXR-induced non-coding RNA (*Fincor*).

• Supplementary file 5. Primer sequences used in this study.

• MDAR checklist

## Data availability

All data generated or analyzed during this study are included in the manuscript and supporting files; RNA-seq and Gro-seq data were deposited in GEO under the accession number GSE221986.

The following dataset was generated:

| Author(s) | Year | Dataset title | Dataset URL | Database and Identifier |
| --- | --- | --- | --- | --- |
| Chen J, Wang R, Xiong F, Sun H, Kemper B, Li W, Kemper JK | 2024 | An FXR-induced novel enhancer RNA, Fincor | https://www.ncbi.nlm.nih.gov/geo/query/acc.cgi?acc=GSE221986 | NCBI Gene Expression Omnibus, GSE221986 |

The following previously published datasets were used:

| Author(s) | Year | Dataset title | Dataset URL | Database and Identifier |
| --- | --- | --- | --- | --- |
| Boergesen M, Gross B, van Heeringen SJ, Hagenbeek D, Bindesbøll C, Caron S, Lalloyer F, Steffensen KR, Nebb HI, Stunnenberg HG, Staels B, Mandrup S | 2012 | Genome-wide profiling of LXR, RXR and PPARα in mouse liver reveals extensive sharing of binding sites | https://www.ncbi.nlm.nih.gov/geo/query/acc.cgi?acc=GSE35262 | NCBI Gene Expression Omnibus, GSE35262 |
| Iwafuchi-Doi M, Donahue G, Kakumanu A, Watts JA, Mahony S, Pugh BF, Lee D, Kaestner KH, Zaret KS | 2016 | Pioneer transcription factor FoxA maintains an accessible nucleosome configuration at enhancers for tissue-specific gene activation [ChIP-seq] | https://www.ncbi.nlm.nih.gov/geo/query/acc.cgi?acc=GSE77670 | NCBI Gene Expression Omnibus, GSE77670 |
| ENCODE consortium | 2011 | H3K27ac ChIP-Seq | https://www.encodeproject.org/files/ENCFF001KMI/ | ENCODE, ENCFF001KMI |

*Continued on next page*

*Continued*

| Author(s) | Year | Dataset title | Dataset URL | Database and Identifier |
|---|---|---|---|---|
| ENCODE consortium | 2011 | H3K4me1 ChIP-Seq | https://www.encodeproject.org/files/ENCFF001KNF/ | ENCODE, ENCFF001KNF |
| Thomas AM, Hart SN, Kong B, Fang J, Zhong XB, Guo GL | 2010 | FXR liver ChIP-Seq | https://genome.ucsc.edu/goldenPath/customTracks/custTracks.html | UCSC, mouse genome |

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
