## [Editor Report · eLife assessment]

Using unbiased transcriptional profiling, the study reports a **fundamental** discovery of FincoR, a novel hepatic lncRNA generated from an enhancer element, which plays a role in FXR biology. The **convincing** findings have therapeutic implications in the treatment of MASH. The authors use state-of-the-art methodology and use unbiased transcriptomic profiling and epigenetic profiling, including validation in mouse models and human samples.

---

## [Referee Report · Reviewer #1 (Public review)]

Summary:

In their article, the authors delve into the therapeutic potential of a newly identified liver-specific lncRNA, FincoR, regulated by the Farnesoid X Receptor (FXR) and induced by the agonist tropifexor, in treating nonalcoholic steatohepatitis (NASH). They demonstrate that FincoR significantly enhances tropifexor's effectiveness in reducing liver fibrosis and inflammation in NASH, presenting it as a promising therapeutic target. The manuscript revisions broaden the study to include both mouse and human data, showing elevated FincoR levels in various mouse models of liver disease and identifying a similar lncRNA in humans, potentially indicating a conserved therapeutic mechanism. This research offers valuable insights into FincoR's role in NASH and suggests further exploration into its functions and mechanisms in liver disease treatment.

Strengths:

This study enhances our understanding of FincoR, a liver-specific lncRNA, and its therapeutic potential in treating NASH through a multifaceted research approach. The revised manuscript further strengthens this contribution by incorporating additional experiments and human relevance, summarized as follows: (1) The use of GRO-seq and RNA-seq technologies has provided an in-depth and unbiased view of the transcriptional alterations driven by the FXR agonist tropifexor, especially emphasizing FincoR's pivotal role. (2) The research expands on the original findings by including diverse mouse models of NAFLD/NASH and cholestatic liver injury. These models demonstrate significant increases in hepatic FincoR levels across various conditions, such as diets high in fat and fructose, chemical induction of liver cholestasis with ANIT, and surgical induction via bile duct ligation. This broadened scope underscores FincoR's involvement in liver disease mechanisms beyond the initial models of FXR knockout (KO) and FincoR liver-specific knockdown (FincoR-LKD). (3) Incorporation of tropifexor, an investigational FXR agonist in clinical trials, alongside these experimental models bridges experimental findings to potential therapeutic applications for NASH patients. (2) The manuscript revision includes promising data on the sequence similarity between mouse FincoR and a human locus, identifying a partially conserved human lncRNA (XR_007061585.1) with elevated levels in NAFLD and PBC patients. This addition enhances the study's relevance to human health. (3) The study's design, with the inclusion of both negative and positive controls and now enriched with a wider array of mouse models and human data, ensures that the observed therapeutic effects can be confidently attributed to FincoR's modulation by tropifexor.

Weaknesses:

The authors acknowledge that certain questions remain unanswered within the scope of this study on FincoR, due to feasibility and technical challenges. While it's important to note that such limitations are rooted in the practical and technical complexities, these unresolved issues might limit the study's immediate impact. The decision to focus on the discovery and initial characterization of FincoR, is strategically but not scientifically justified.

---

## [Referee Report · Reviewer #2 (Public Review)]

Summary:

Nonalcoholic fatty liver disease (NASH), recently renamed as metabolic dysfunction-associated steatohepatitis (MASH) is a leading cause of liver-related death. Farnesoid X receptor (FXR) is a promising drug target for treating NASH and several drugs targeting FXR is under clinical investigation for its efficacy in treating NASH. The authors intended to address whether FXR mediates its hepatic protective effects through regulation of lncRNAs, which would provide novel insights into the pharmacological targeting of FXR for NASH treatment. The authors went from an unbiased transcriptomics profiling to identify a novel enhancer-derived lncRNA FincoR enriched in the liver and showed that the knockdown of FincoR in a murine NASH model attenuated part of the effect of tropifexor, an FXR agonist, namely inflammation and fibrosis, but not steatosis. This study provides a framework how one can investigate the role of noncoding genes in pharmacological intervention targeting a known protein coding genes. Given that many disease-associated genetic variants are located in the non-coding regions, this study, together with others, may provide useful information for improved and individualized treatment for metabolic disorders.

Strengths:

The study leverages both transcriptional profile and epigenetic signatures to identify the top candidate eRNA for further study. The subsequent biochemical characterization of FincoR using FXR-KO mice combined with Gro-seq and Luciferase reporter assays convincingly demonstrates this eRNA as a FXR transcriptional targets sensitive to FXR agonists. The use of in vitro culture cells and the in vivo mouse model of NASH provide multi-level evaluation of the context-dependent importance of the FincoR downstream of FXR in regulation of functions related to liver dysfunction.

Weaknesses:

Future work to dissect the detailed mechanisms by which FincoR facilitates action of FXR and its agonists is warranted. A more direct approach to alter eRNA levels, e.g., overexpression of FincoR in the liver would provide important data to interpret its functional regulation.

---

## [Author Response]

The following is the authors’ response to the original reviews.

**eLife assessment:**
The authors report a novel hepatic lncRNA FincoR regulated by FXR with therapeutic implications in the treatment of MASH. The findings are important and use an appropriate methodology in line with the current state-of-the-art, with convincing support for the claims.
**Public Reviews:**

**Reviewer #1 (Public Review):**
Summary:In the article titled "Hammerhead-type FXR agonists induce an eRNA FincoR that ameliorates nonalcoholic steatohepatitis in mice," the authors explore the role of the Farnesoid X Receptor (FXR) in treating metabolic disorders like NASH. They identify a new liver-specific long non-coding RNA (lncRNA), FincoR, regulated by FXR, notably induced by agonists such as tropifexor. The study shows that FincoR plays a significant role in enhancing the efficacy of tropifexor in mitigating liver fibrosis and inflammation associated with NASH, suggesting its potential as a novel therapeutic target. The study makes a promising contribution to understanding the role of FincoR in alleviating liver fibrosis in NASH, providing initial insights into the mechanisms involved. While it offers a valuable starting point, there is potential for further exploration into the functional roles of FincoR and their specific actions in human NASH cases. Building upon the current findings to elucidate more detailed mechanistic pathways through which FincoR exerts its therapeutic effects in liver disease would elevate the research's significance and potential impact in the field.Strengths:This study stands out for its comprehensive and unbiased approach to investigating the role of FincoR, a liver-specific lncRNA, in the treatment of NASH. Key strengths include: (1) The application of advanced sequencing methods like GRO-seq and RNA-seq offered a comprehensive and unbiased view of the transcriptional changes induced by tropifexor, particularly highlighting the role of FincoR. (2) Utilizing a genetic mouse model of FXR KO and a FincoR liver-specific knockdown (FincoR-LKD) mouse model provided a controlled and relevant environment for studying NASH, allowing for precise assessment of tropifexor's therapeutic effects. (3) The inclusion of tropifexor, an investigational new drug in clinical trials, adds significant clinical relevance to the study. It bridges the gap between experimental research and potential therapeutic application, providing a direct pathway for translating these findings into real-world clinical benefits for NASH patients. (4) The study's rigorous experimental design, incorporating both negative and positive controls, ensured that the results were specifically attributable to the action of FincoR and tropifexor.Weaknesses:The study presents several notable weaknesses that could be addressed to strengthen its findings and conclusions: (1) The authors focus on FincoR, but do not extensively test other lncRNAs identified in Figure 1A. A more comprehensive approach, such as rescue experiments with these lncRNAs, would provide a better understanding of whether similar roles are played by other lncRNAs in mitigating NASH. (2) FincoR was chosen for further study primarily because it is the most upregulated lncRNA induced by GW4064. Including another GW4064-induced lncRNA as a control in functional studies would strengthen the argument for FincoR's unique role in NASH. (3) The study does not conclusively demonstrate whether FincoR is specifically expressed in hepatocytes or other liver cell types. Conducting FincoR RNA-FISH with immunofluorescent experiments or RT-PCR, using markers for different liver cell types, would clarify its expression profile. (4) Understanding the absolute copy number of FincoR is crucial. Determining whether there are sufficient copies of FincoR to function as proposed would lend more credibility to its suggested role. (5) The manuscript, although technically proficient, does not thoroughly address the relevance of these findings to human NASH. Questions like the conservation of FincoR in humans and its potential role in human NASH should be discussed.
**Reviewer #2 (Public Review):**
Summary:Nonalcoholic fatty liver disease (NASH), recently renamed as metabolic dysfunctionassociated steatohepatitis (MASH) is a leading cause of liver-related death. Farnesoid X receptor (FXR) is a promising drug target for treating NASH and several drugs targeting FXR are under clinical investigation for their efficacy in treating NASH. The authors intended to address whether FXR mediates its hepatic protective effects through the regulation of lncRNAs, which would provide novel insights into the pharmacological targeting of FXR for NASH treatment. The authors went from an unbiased transcriptomics profiling to identify a novel enhancer-derived lncRNA FincoR enriched in the liver and showed that the knockdown of FincoR in a murine NASH model attenuated part of the effect of tropifexor, an FXR agonist, namely inflammation and fibrosis, but not steatosis. This study provides a framework for how one can investigate the role of noncoding genes in pharmacological intervention targeting known protein-coding genes. Given that many disease-associated genetic variants are located in the non-coding regions, this study, together with others, may provide useful information for improved and individualized treatment for metabolic disorders.Strengths:The study leverages both transcriptional profile and epigenetic signatures to identify the top candidate eRNA for further study. The subsequent biochemical characterization of FincoR using FXR-KO mice combined with Gro-seq and Luciferase reporter assays convincingly demonstrates this eRNA as a FXR transcriptional target sensitive to FXR agonists. The use of in vitro culture cells and the in vivo mouse model of NASH provide multi-level evaluation of the context-dependent importance of the FincoR downstream of FXR in the regulation of functions related to liver dysfunction.Weaknesses:As discussed, future work to dissect the mechanisms by which FincoR facilitates the action of FXR and its agonists is warranted. It would be helpful if the authors could base this on the current understanding of eRNA modes of action and the observed biochemical features of FincoR to speculate potential molecular mechanisms explaining the observed functional phenotype. It is unclear if this eRNA is conserved in humans in any way, which will provide relevance to human disease. Additionally, the eRNA knockdown was achieved by deletion of an upstream region of the eRNA transcription. A more direct approach to alter eRNA levels, e.g., overexpression of FincoR in the liver would provide important data to interpret its functional regulation.

We thank the Editor and Reviewers for their constructive comments. We believe we have addressed all of the issues (detailed below) and the revisions have greatly strengthened the manuscript.

**Reviewer 1:**
The study presents several notable weaknesses that could be addressed to strengthen its findings and conclusions:(1) The authors focus on FincoR, but do not extensively test other lncRNAs identified in Figure 1A. A more comprehensive approach, such as rescue experiments with these lncRNAs, would provide a better understanding of whether similar roles are played by other lncRNAs in mitigating NASH.(2) FincoR was chosen for further study primarily because it is the most upregulated lncRNA induced by GW4064. Including another GW4064-induced lncRNA as a control in functional studies would strengthen the argument for FincoR's unique role in NASH.(3) The study does not conclusively demonstrate whether FincoR is specifically expressed in hepatocytes or other liver cell types. Conducting FincoR RNA-FISH with immunofluorescent experiments or RT-PCR, using markers for different liver cell types, would clarify its expression profile.(4) Understanding the absolute copy number of FincoR is crucial. Determining whether there are sufficient copies of FincoR to function as proposed would lend more credibility to its suggested role.

Response to 1 - 4: We thank Reviewer 1 for the positive comments on the strength of our work, including the open-ended approach, the novel eRNA FincoR and its strong relevance to liver disease. We also value the constructive feedback provided by the reviewer and agree that additional studies are important to fully understand the mechanisms of FincoR and the functional significance of other FXR-induced lncRNAs. In this manuscript we report the discovery and initial characterization of FincoR, as well as its potential function in FXR action in response to hammerhead agonists, but a number of interesting questions are raised. Future experiments, as suggested by reviewer, will be needed to examine the role of other FXR-induced lncRNAs, the potential role of FincoR induction by other nuclear receptors with binding sites at FincoR, whether FincoR is expressed in liver cell types in addition to hepatocytes, and the expression abundance of FincoR. These are all excellent suggestions for future experimentation which we feel are beyond the scope of the present report. For example, generating a genetic CRISPR/Cas9 of another lncRNA is not trial as it takes a significant amount of work with murine models. Also, we did not mean to exclude if other lncRNAs induced by FXR also bear functions. Technically, rescue experiment is not possible as FincoR RNA can be potentially very long (~10 kb if estimated by RNA-seq pattern in Fig.1C), and it is not feasible now to properly express it by exogenous vectors to ensure the expression levels are similar to endogenous ones. We therefore consider that these important questions are more suitable for future work to fully address. Our belief is that a comprehensive exploration of FXR-regulated lncRNAs holds the potential to unveil novel insights crucial for the development of therapies targeting NASH and other metabolic diseases. The study of FincoR is the beginning of this area of research.

(5) The manuscript, although technically proficient, does not thoroughly address the relevance of these findings to human NASH. Questions like the conservation of FincoR in humans and its potential role in human NASH should be discussed.

Response: These are important questions. To respond to the reviewer’s comment, new experiments are presented in our final revised manuscript in which we utilized mouse models of NAFLD/NASH and cholestatic liver injury to determine FincoR’s role in these diseases. Hepatic FincoR levels were significantly increased in mice fed with high fat diet (HFD) for 12 weeks (Figure 8A) and in mice fed a HFD with high fructose (HFHF) in drinking water for 12 weeks (Figure 8B). Elevated hepatic FincoR levels were also observed in mice treated with α-naphthylisothiocyanate (ANIT), a chemical inducer of liver cholestasis (Figure 8C), and in mice with bile duct ligation (BDL), a surgical method to induce cholestatic liver injury (Figure 8D).

In terms of the human relevance, we have provided additional information and figures showing that there is sequence similarity between mouse FincoR and a human loci. FincoR sequence is moderately conserved between mice and humans as displayed in the UCSC genome browser (Figure 8E). Annotation of these conserved human sequences revealed that they overlap with a functionally uncharacterized human lncRNA XR_007061585.1 (Figure 8F). Further, we conducted qRT-PCR experiment from human patient’s RNA samples, which demonstrated that hepatic lncRNA XR_007061585.1 levels are elevated in patients with NAFLD and PBC, but not in severe NASH-fibrosis patients (Figure 8G-H). These results demonstrate that hepatic levels of a potential human analog of FincoR are elevated in NAFLD and PBC patients, which is consistent with FincoR’s upregulation in mouse models of chronic liver disease with hepatic inflammation and liver injury. Whether human lncRNA XR_007061585.1 is entirely analogous to mouse FincoR in terms of functions and mechanisms, and whether the elevation of this human lncRNA hasa role in liver disease progression or is an adaptive response to liver injury remains to be determined.

**Reviewer #2 (Recommendations For The Authors):**
(1) In the introduction Line 96, "..., while the vast majority are transcribed into ncRNAs" may not be accurate. Please refer to Pointing and Haerty Annu Rev 2022 for a related discussion.

Response: We would like to thank the reviewer for pointing out this inaccurate information in the introduction. We have changed the content in the text, “While a significant portion of the genome was initially thought to be "junk DNA", it has been established that many non-coding regions give rise to functional non-coding RNAs.”

(2) Figure 5: the authors should provide a clear illustration demonstrating the sequence targeted by the sgRNA in relation to the transcriptional and epigenetic profile (i.e., RNAseq and H3K27ac ChIP-seq data).

Response: The illustration (Figure 5-figure supplement 1A, right panel) demonstrating the sequence targeted by the sgRNA has been updated as suggested by the reviewer.

In this model, the upstream of FincoR is deleted, leading to the inhibition of FincoR transcription. Does the deleted region include FXR binding sites? If so, would the phenotype be due to the deletion of these binding sequences, rather than the decreased FincoR transcripts? Accordingly, the limitation or alternative interpretation should be discussed.

Response: The reviewer made a good point. The deleted region includes FXR binding sites so that we cannot rule out decreased binding of FXR or decreased transcription of the region per se, in addition to the decreased levels of FincoR, to bear a role in the phenotypic changes we observed. In the final revision, we have added discussion of this alternative (6th paragraph in the revised discussion section).

(3) Figure 6C, the images should be accompanied by quantification. It appears the FincoR-KD shows a visible difference as compared to Tropifexor-treated control mice, which does not match entirely what is written in the results.

Response: The quantitation of Oil Red O staining has been done as suggested by the reviewer (Figure 6C). The result is consistent with the triglyceride result showing that tropifexor treatment markedly reduced neutral lipids determined by Oil Red O staining of liver sections (Figure 6C) and liver TG levels (Figure 6D) and these beneficial effects on reducing fatty liver were not altered by FincoR.

(4) Figure 7, does AST show the same pattern as ALT? As indicated from Line 335, "tropifexor treatment reduced mRNA levels of several genes that promote fibrosis (Col1a1, Col1a2, ...)". Fig. 7D does not seem to match the description of Col1a1. Authors may need to modify the results.

Response: AST has been measured and has the same pattern as ALT. The new data have been added to Figure 7B. Col1a1 expression has been re-measured and the results have been updated in Figure 7D.

(5) Is FincoR level reduced in NASH conditions?

Response: We thank the Reviewer for this question. We now added new data to examine the levels of FincoR in mouse liver disease models and also examined levels of a potential human analog of FincoR in human liver specimens from PBC, NAFLD, and NASH patients. Please see our new data and description above in the response to comment 5 by Reviewer 1 (most data now included in the new Figure 8).

(6) Please provide information on the conservation of FincoR (DNA and RNA) in humans. This would be important to provide the human disease relevance.

Response: As described above in the response to comment 5 of reviewer 1, a human loci shows sequence similarity to mouse FincoR and this conserved region has an annotated uncharacterized human lncRNA. We also examined the levels of this human homolog in human diseased liver samples. Our new results demonstrate that hepatic levels of a potential human analog of FincoR are elevated in NAFLD and PBC patients, which is consistent with FincoR’s upregulation in mouse models of chronic liver disease with hepatic inflammation and liver injury. Whether human lncRNA XR_007061585.1 is entirely analogous to mouse FincoR in terms of functions and mechanisms, and whether the elevation of this human lncRNA has a role in liver disease progression or is an adaptive response to liver injury remains to be determined.

(7) Several discussion points for the authors' consideration:(7.1) human-mouse conservation as alluded to in #6;

Response: Potential human-mouse conservation is discussed with new data in the last paragraph of the Results section.

(7.2) potential molecular mechanism involved in FincoR-regulated hepatocyte function;

Response: We thank Reviewer for this comment. We have added more discussion as shown below: “RNA inside the cells usually associates with different RNA-binding proteins (RBPs). To predict those potential binding proteins of FincoR. Additional bioinformatic analysis identified proteins that potentially binding FincoR, including KHDRBS1, RBM38, YBX2 and YBX3 (Supplemental file 4). These findings and potential functions of the binding proteins are discussed in the 5th paragraph of the discussion section in the final revised manuscript. Whether these predicted RBPs interact with FincoR and the underlying mechanisms will need to be investigated in future experimentation to understand the mechanisms involved in FincoR-regulated hepatocyte function.”

(7.3) any disease-associated SNPs in the FincoR locus.

Response: No SNPs were noted in the annotation of the human loci with sequence similarity to mouse FincoR in the NCBI genome data viewer.

(7.4) the in vitro induction of FincoR is transient but in vivo this occurs after 12 days of drug treatment. How do the authors reconcile the differential induction patterns?

Response: To clarify, the induction of FincoR after a single dose of GW4064 in vivo was transient, peaked within 1 h and then declined gradually (Figure 1-figure Supplement 1C). In the tropifexor treatment protocol (also in vivo), the mice were treated daily with tropifexor for 12 days so that the multiple doses maintained FincoR induction. The beneficial effect of tropifexor by inducing FincoR, therefore, accumulated over the 12 days.

It is worthy to note that we failed to see induction of FincoR in isolated primary mouse hepatocytes treated with GW4064 in vitro. We can only detect FincoR in primary hepatocytes isolated from GW4064-treated mice liver. This may be due to the loss of key factors mediating FincoR induction in the cultured primary hepatocytes.